# Non-Perturbative Propagators in Quantum Gravity

**Benjamin Knorr** [1,*] and **Marc Schiffer** [2]

1   Perimeter Institute for Theoretical Physics, Waterloo, ON N2L 2Y5, Canada
2   Institut für Theoretische Physik, Universität Heidelberg, 69120 Heidelberg, Germany;
    schiffer@thphys.uni-heidelberg.de
*   Correspondence: bknorr@perimeterinstitute.ca

**Abstract:** We employ non-perturbative renormalisation group methods to compute the full momentum dependence of propagators in quantum gravity in general dimensions. We disentangle all different graviton and Faddeev–Popov ghost modes and find qualitative differences in the momentum dependence of their propagators. This allows us to reconstruct the form factors that are quadratic in curvature from first principles, which enter physical observables like scattering cross sections. The results are qualitatively stable under variations of the gauge fixing choice.

**Keywords:** quantum gravity; functional renormalisation; momentum dependence; propagator

## 1. Introduction

The unification of gravity with quantum mechanics is a notoriously hard problem in theoretical physics. Even a century after the development of quantum theory and general relativity, breakthroughs in the understanding of quantum properties of spacetime are few and far between. This is not least because of the typical scale that we expect quantum gravity effects to be important at—the Planck scale, which is about $10^{-35}$ m. To illustrate this fantastically small number, measuring the typical size of a human to an accuracy of a Planck length is roughly comparable to measuring the extension of the Milky Way with an accuracy of an atomic nucleus. This emphasises why experimental data on quantum gravity effects are hard to come by and, accordingly, quantum gravity theories presently mostly rely on theoretical considerations, and can only be confronted with consistency tests.

A conservative strategy, and potentially a path to success in formulating a theory of quantum gravity, is to build on well-established theories that are valid at larger length scales, and only add as few extra assumptions as possible to extend the theory to include new phenomena. In the context of quantum gravity, such an approach is the Asymptotic Safety program [1]. It embraces the importance of symmetries in the understanding of the Standard Model and adds an interacting ultraviolet completion of gravity in the form of a quantum realisation of scale symmetry. An enormous advantage of this approach is its closeness to standard quantum field theory notions, and the straightforward connection to low energy physics. In the matter sector, especially, this allows us to confront the scenario with many observational consistency tests. A disadvantage is that the existence of such a scale invariant regime is hard to prove. In practice, however, this is less of a problem. Many interacting fixed points have been found with satisfactory precision in other contexts, for example, in statistical physics and condensed matter systems [2–5]. By now, there is ample evidence that a suitable fixed point indeed exists not only in various approximations on the dynamics of pure quantum gravity [6–48], but also in the presence of matter [49–76]. Phenomenological implications of the interacting fixed point have been discussed in the context of particle physics [50,61,70,71,77–85], cosmology [40,86–93], and black holes [94–102]. For recent reviews and introductions, see [103–107], and for a critical discussion of the status of the field, see [108,109]. Evidence for asymptotic safety in gravity has also been found in lattice formulations of quantum gravity, namely

Euclidean and Causal Dynamical Triangulations [110–118]. On the lattice, a scale invariant regime can be realised if a second-order phase transition exists that gives rise to consistent low-energy physics.

The physical renormalisation group running of couplings is one of the key objects of study in quantum field theories. This translates into the momentum dependence of correlation functions, which are the basic building blocks of observables such as scattering cross sections. The easiest non-trivial correlation function is the propagator, which is the inverse of the two-point correlation function. It stores important information about the unitarity and causality of the theory as it is related to the spectral function (if it exists), see, for example, [119]. An accurate description of the propagator is thus crucial for deciding whether Asymptotic Safety provides a theory of quantum gravity that is compatible with these notions.

In this work, we compute the full momentum dependence of the propagators of the graviton and its accompanying Faddeev–Popov ghost with non-perturbative renormalisation group techniques. A key advance is that we distinguish between all the different modes of the graviton and the ghost. As is well-known from representation theory, the graviton splits into a gauge-invariant spin two mode (the transverse-traceless mode), a gauge-invariant spin zero mode, and a pure gauge vector mode (which can be split into a spin one transverse, and a spin zero longitudinal component). The ghost is a vector field itself, and also splits into a transverse and a longitudinal mode. The physical information is stored in the spin two and gauge-invariant spin zero mode. The momentum dependence of their propagators can be mapped to the form factors, which are quadratic in curvature tensors.

The central results can be summarised as follows:

- The spin two and spin zero modes of the graviton feature qualitative and quantitative differences in their momentum dependence.
- The overall gauge and gap dependence is small, see Figures 4 and 5.
- Within our approximation, only the spin two mode shows the property of momentum locality [31], and furthermore only in four dimensions and with equal three- and four-graviton coupling, see (65).
- A derivative expansion of the form factors shows alternating signs, indicating that computations relying on such an expansion can be inherently unstable and generically introduce fictitious poles [120], see (85).
- The propagators of the two ghost modes are related for asymptotic momenta, but differ for finite momenta, see (60) and (70).
- Quantum corrections to the free propagator decrease exponentially with increasing dimension, see (78) and Figure 3.

This work is structured as follows: our renormalisation group tool of choice is the so-called functional renormalisation group, which we briefly review in Section 2. In Section 3 we discuss our setup and some general aspects of momentum-dependent correlation functions. Section 4 and Appendix A comprise a discussion of analytical properties of the momentum dependence of the propagators, whereas in Section 5, we discuss the numerical results. Sections 6 and 7, as well as Appendix B, contain a comparison of results obtained in different approximation schemes. Finally, we conclude with a summary and an outlook in Section 8.

## 2. Functional Renormalisation Group

To extract the scale dependence of the system, we employ the functional renormalisation group (FRG). It is based on the Wetterich equation [121–123], describing the flow of the scale-dependent action $\Gamma_k$,

$$\partial_t \Gamma_k[h, \bar{g}] = \frac{1}{2} \mathrm{STr}\left[ \left( \Gamma_k^{(2)}[h, \bar{g}] + \mathfrak{R}_k[\bar{g}] \right)^{-1} \partial_t \mathfrak{R}_k[\bar{g}] \right], \tag{1}$$

which was adapted to gravity in [6]. Here, we denote $t = \log k$ as the "RG-time", $\Gamma_k^{(2)}$ is the second functional derivative of $\Gamma_k$ with respect to fluctuation fields, and the super-trace STr contains a sum over all fields and indices, as well as an integration over the continuous coordinates. The regulator $\mathfrak{R}_k$, which enters the generating functional as a momentum-dependent mass term for the fluctuation, provides an infrared (IR) regularisation of modes with $p^2 < k^2$, and thereby ensures IR finiteness. The factor $\partial_t \mathfrak{R}_k$ ensures ultraviolet (UV) finiteness by cutting off modes with $p^2 > k^2$. Overall, the regulator term, together with its derivative, implement the Wilsonian idea of momentum-shell wise integration of quantum fluctuations. As a result, the scale-dependent effective action $\Gamma_k$ interpolates between the classical action $S$ when no quantum fluctuations are integrated out, that is, in the limit $k \to \infty$, and the full quantum effective action $\Gamma$, when $k \to 0$. A central role is played by the fixed points of the flow, which realise quantum scale invariance, where all couplings measured in units of the scale $k$ are scale-independent. For reviews of the FRG, see [106,124–128]. Due to the momentum-shell wise integration of quantum fluctuations, the Wetterich Equation (1) is formulated on Euclidean backgrounds, since only then does the squared momentum provide definite information on whether a certain mode is a UV or an IR mode. In the following, we will therefore assume a Euclidean background and hence discuss Euclidean quantum gravity. The generalisation of the functional RG to Lorentzian spacetimes is one of the important challenges of the approach [109], and first steps towards investigations of Lorentzian spacetimes have been presented in [20,23,37,129–134].

Due to the formulation as a local coarse graining and the necessity of a gauge fixing, the formal introduction of a background metric $\bar{g}$ is hard to avoid. In this spirit, the full metric $g$ is split into background metric $\bar{g}$ and a (not necessarily small) metric perturbation $h$ according to

$$g_{\mu\nu} = \bar{g}_{\mu\nu} + h_{\mu\nu}. \tag{2}$$

Other parameterisations have been investigated, for example, in [33–35,53,63,129,135–145]. The practical advantage of the background field method is that one can retain background diffeomorphism invariance at each step. Within the FRG, the background metric $\bar{g}$ only serves as a technical tool to allow for a momentum-shell wise integration of quantum fluctuations, and in principle never has to be specified. Without approximations of the dynamics of the theory, the physical results obtained at $k \to 0$ are therefore entirely independent of the specific choice of $\bar{g}$. However, in the following sections, we will use a concrete choice for $\bar{g}$, since this significantly reduces the computational complexity.

Since the scale-dependent effective action $\Gamma_k$ contains all operators that are consistent with the symmetries of the system, practical computations of RG flows of couplings require the truncation of $\Gamma_k$ to a manageable set of operators. In gauge theories, especially, the momentum dependence of scale-dependent correlation functions provides crucial information about the system, such as on unitarity. Thus, in approximations, the resolution of momentum dependence is unavoidable to obtain reliable results. A scale identification, which allows us to translate the RG running of operators into their momentum dependence is, in general, insufficient. For example, higher order $n$-point correlation functions generically depend on $n$ different momenta independently, such that the momentum dependence cannot be captured by the dependence on just one scale $k$. It is, however, possible that, for special momentum configurations, the RG running qualitatively agrees with the physical momentum dependence [119]. Furthermore, the RG scale $k$ is an artificial scale introduced via the regulator $\mathfrak{R}_k$ to regularise the path integral. Physical quantities, such as scattering amplitudes, can only be extracted in the limit $k \to 0$, where all contributions from the regulator $\mathfrak{R}_k$ vanish. In this limit, the UV behaviour of a given theory is described by the momentum dependence of operators. The momentum-dependent evaluation of $\beta$-functions is thus necessary to extract the UV physics of a system.

In the context of asymptotically safe quantum gravity, different expansion schemes are used, which allow us to extract momentum-dependent flow equations. On the one hand, a vertex expansion in terms of metric fluctuations $h$ is employed [24,29]. In this so-called fluctuation approach, see [107] for a review, the starting point is a seed action $S$, which is then

expanded in terms of fluctuation fields $h$, typically around a flat background, see [43,69] for expansions around non-flat backgrounds. Consequently, the different vertices are labelled with scale-dependent couplings, the flow of which is evaluated for different values of the external momenta, allowing us to extract momentum-dependent flows of $n$-point correlation functions. In this approach, the momentum-dependent RG running of vertex correlation functions of the graviton two- [29], three- [31] and four-point functions [39], as well as three-point functions in gravity-matter systems [52,58,64–66] have been investigated. As an important approximation, all computations in the fluctuation approach identify the scale dependence of different tensor structures of the graviton with the scale dependence of the purely transverse–traceless part of the corresponding $n$-point correlator. On the other hand, the form factor expansion [68,101] is based on diffeomorphism invariant operators, where the corresponding form factors are general functions of covariant derivatives. In this expansion, the scale dependence of the entire form factor can be extracted within the background field approximation. As a caveat, the background field approximation ignores that the regulator and the gauge fixing break the full diffeomorphism symmetry, which causes the scale-dependent effective action $\Gamma_k$ to depend on $\bar{g}$ and $h$ individually [15,16,24,29–31,39,42,43,52,64–66,68,69,107,119,144,146–163]. Comparing fluctuation and background results at the same level of approximation thus provides information about how much the respective modified Ward identities are broken.

In the present work, we provide a comparison of both expansion schemes on the level of the two-point function. For this purpose, we will set the stage and provide more details on momentum dependence in Section 3. In Sections 4 and 5, we derive and analyse the momentum-dependent flow equation for the graviton two-point correlator for general dimension $d$. Importantly, we distinguish all different tensor structures at the level of the two-point function. After gauge fixing, we compute the scale dependence of four different tensor structures for the graviton and ghost two-point functions. Additionally, in Section 6, we compute the corresponding form factors on the background field level, which contribute to the gravitational two-point function. We show that on this level in the expansion and under certain assumptions, there is a one-to-one correspondence between vertex correlation functions and form factors. Furthermore, in Section 7, we compare both expansions based on the analysis of asymptotic behaviours of form factors and correlation functions.

## 3. Momentum Dependence in Quantum Gravity

In this section, we discuss our setup to resolve momentum-dependent correlation functions. We start with a discussion of the correlators themselves. Next, we present the general structure of the RG flow. Finally, we relate the form factor expansion to the fluctuation correlation functions.

### 3.1. Fluctuation Approach

For the computation of vertex correlation functions, we choose a flat Euclidean background

$$\bar{g}_{\mu\nu} = \delta_{\mu\nu}\,, \tag{3}$$

which is a technical choice simplifying the computations significantly. In principle, each diffeomorphism invariant operator in the seed action is labelled by a single coupling. However, in the presence of the regulator and gauge fixing, the scale dependence of different $n$-point functions, originating from the same operator in the seed action, generally differ [39,58,64–66]. Therefore, we will introduce separate couplings for each operator in the vertex expansion. Furthermore, on the level of the two-point function, we will distinguish between the transverse–traceless (TT) mode, and the gauge-invariant scalar mode. Specifically, the seed action we will use in the following reads

$$S = S_{\text{grav}} + S_{\text{gh}} + S_{\text{gf}}\,, \tag{4}$$

where we approximate $S_{\text{grav}}$ by the Einstein–Hilbert action $S_{\text{EH}}$ describing classical gravity:

$$S_{\text{EH}} = -\frac{1}{16\pi G_N} \int \mathrm{d}^d x \sqrt{g}(R - 2\Lambda) \,. \tag{5}$$

Here, $G_N$ and $\Lambda$ are the Newton coupling and the cosmological constant, respectively. Computing the effect of quantum fluctuations of gravity requires the inclusion of a gauge fixing condition $F^\mu = 0$, where we choose

$$F^\mu = \left( \bar{g}^{\mu\kappa} \bar{D}^\lambda - \frac{1 + \beta_h}{d} \bar{g}^{\kappa\lambda} \bar{D}^\mu + \frac{\gamma_h}{d} \bar{D}^\mu \bar{D}^\kappa \frac{1}{\bar{D}^2} \bar{D}^\lambda \right) h_{\kappa\lambda} \,. \tag{6}$$

Here, $\bar{D}^\mu$ refers to the background covariant derivative, and $\beta_h$ and $\gamma_h$ are gauge parameters. The presence of $\gamma_h$ allows us to compute the scale dependence of all tensor structures on the level of the graviton two-point function. The gauge fixing condition is implemented into the action via the inclusion of the gauge fixing action

$$S_{\text{gf}} = \frac{1}{32\pi \alpha_h G_N} \int \mathrm{d}^d x \sqrt{\bar{g}}\, F^\mu \bar{g}_{\mu\nu} F^\nu \,. \tag{7}$$

The gauge parameter $\alpha_h$ controls how strongly we implement the gauge fixing condition in the path integral. A strict implementation corresponds to the Landau limit $\alpha_h \to 0$. The resulting Faddeev–Popov determinant is taken care of by introducing ghost fields $c^\mu$ and $\bar{c}^\nu$, with ghost action $S_{\text{gh}}$:

$$S_{\text{gh}} = \frac{1}{16\pi G_N} \int \mathrm{d}^d x \sqrt{\bar{g}}\, \bar{c}_\mu \frac{\delta F^\mu}{\delta h_{\sigma\kappa}} \mathcal{L}_c g_{\sigma\kappa} \,. \tag{8}$$

The Lie derivative $\mathcal{L}_c g_{\sigma\kappa}$ of the full metric $g_{\mu\nu}$ along the direction of the ghost field is given by

$$\mathcal{L}_c g_{\sigma\kappa} = 2\bar{g}_{\rho(\sigma} \bar{D}_{\kappa)} c^\rho + c^\rho \bar{D}_\rho h_{\sigma\kappa} + 2 h_{\rho(\sigma} \bar{D}_{\kappa)} c^\rho \,. \tag{9}$$

Round brackets indicate a normalised symmetrisation.

With the seed action specified, and using the parameterisation (2), we expand the scale-dependent effective action in powers of the fluctuation fields, via the vertex expansion [24,29]

$$\Gamma_k[\Phi, \bar{g}] = \sum_{n=0}^{\infty} \frac{1}{n!} \Gamma^{(n)}_{k\, A_1 \dots A_n}[0, \bar{g}] \Phi^{A_n} \dots \Phi^{A_1} \,, \tag{10}$$

where we have introduced the superfield $\Phi$ as a collection of all dynamical fields in our system,

$$\Phi^A = \left( h_{\mu\nu}(x), c^\mu(x), \bar{c}_\mu(x) \right) \,. \tag{11}$$

The Einstein summation convention over the superindex A implies the summation over discrete indices as well as an integration over the coordinates. Furthermore, $\Gamma^{(n)}_k$ refers to the $n$-th functional derivative with respect to the superfield $\Phi$. For convenience, we rescale the vertices according to

$$\Gamma^{(n)}_{k\, A_1 \dots A_n}[0, \bar{g}, G_N, \Lambda] \to \left( k^{-2} g_n \right)^{n/2} \Gamma^{(n)}_{k\, A_1 \dots A_n}[0, \bar{g}, k^{-2} g_n, k^2 \lambda_n] \,. \tag{12}$$

As mentioned at the beginning of the section, we have introduced individual dimensionless couplings, $g_n$ and $\lambda_n$, that label the different $n$-point functions. The vertices introduced in (12) also satisfy flow equations similar to (1), see for example, [39]. Due to the breaking of diffeomorphism invariance by the gauge fixing term and the regulator, the scale dependence of these different couplings does not necessarily agree. In addition to the distinction of different dimensionless $n$-point couplings $g_n$ and $\lambda_n$, the couplings of pure gravity and ghost-gravity, or gravity-matter $n$-point vertices will also differ, and

should be distinguished. Therefore, for vertices originating from the ghost action (8), one generally has to replace the coupling $g_n$ in (12) by $g_c$. For the course of this work, we will however work under the assumption that the differences between those couplings, caused by the breaking of diffeomorphism invariance, are negligible, and therefore assume that $g_n = g_c = g$, and $\lambda_n = \lambda$, for $n \geq 3$, and only distinguish the two-point functions from higher-order vertices. In this way, we assume the exact realisation of *effective universality* [39,64–66], which refers to the semi-quantitative agreement of the scale dependence of different $n$-point vertices.

In the present work, we will focus on the two-point function of the pure gravity system. For the graviton two-point function, there are five independent tensor structures, three of them associated with the gauge fixing parameters $\alpha_h$, $\beta_h$ and $\gamma_h$. The two remaining tensor structures, which will be unaffected by the choice of gauge, are related to the transverse–traceless mode $h^{\text{TT}}_{\mu\nu}$, which satisfies

$$\bar{D}^\mu h^{\text{TT}}_{\mu\nu} = 0, \qquad \bar{g}^{\mu\nu} h^{\text{TT}}_{\mu\nu} = 0, \tag{13}$$

and the scalar mode $h_0$ defined as

$$h^0_{\mu\nu} = \Pi^0_{\ \mu\nu}{}^{\alpha\beta} h_{\alpha\beta}, \tag{14}$$

where the scalar projector $\Pi^0$ is orthogonal to the gauge fixing action and the transverse-traceless projector, that is,

$$\Pi^0 \cdot S^{(2)}_{\text{gf}} = S^{(2)}_{\text{gf}} \cdot \Pi^0 = 0, \qquad \Pi^0 \cdot \Pi^{\text{TT}} = 0. \tag{15}$$

Explicitly, the transverse–traceless projector in momentum space is given by

$$\begin{aligned}
\Pi^{\text{TT}}_{\ \mu\nu}{}^{\rho\sigma} &= \delta_{(\mu}^{\ \rho} \delta_{\nu)}^{\ \sigma} - \frac{1}{d-1} \bar{g}_{\mu\nu} \bar{g}^{\rho\sigma} - \frac{2}{p^2} \delta_{(\mu}^{\ (\rho} p_{\nu)} p^{\sigma)} \\
&\quad + \frac{1}{d-1} \frac{1}{p^2} \left( \bar{g}_{\mu\nu} p^\rho p^\sigma + p_\mu p_\nu \bar{g}^{\rho\sigma} \right) + \frac{d-2}{d-1} \frac{1}{p^4} p_\mu p_\nu p^\rho p^\sigma.
\end{aligned} \tag{16}$$

The projector on the scalar mode $h_0$, referred to in (15), reads

$$\Pi^0_{\ \mu\nu}{}^{\rho\sigma} = \frac{B^2}{C} \left( \bar{g}_{\mu\nu} + \frac{A}{B} \frac{p_\mu p_\nu}{p^2} \right) \left( \bar{g}^{\rho\sigma} + \frac{A}{B} \frac{p^\rho p^\sigma}{p^2} \right), \tag{17}$$

with

$$A = (d\beta_h - \gamma_h), \tag{18}$$
$$B = (d - \beta_h - 1 + \gamma_h), \tag{19}$$
$$C = (d-1) \left( \gamma_h(-2\beta_h + \gamma_h - 2) + d^2 + d\left( \beta_h^2 + 2\gamma_h - 1 \right) \right). \tag{20}$$

The projectors $\Pi^{\text{TT}}$ and $\Pi^0$ can also be formulated in curved spaces, see for example, [48]. We will now discuss how we resolve the two-point function. We start with the momentum-independent parts of the correlator. We can introduce two gaps, $\mu_{\text{TL}}$ and $\mu_0$, which correspond to the traceless and the trace sector, respectively. These gaps $\mu_x$ are related to the different $n$-point couplings introduced in (12) via $\mu_x = -2\lambda_{2,x}$, where the subscript $x$ labels the respective mode. Explicitly, we introduce them by adjusting the two-point function via [42]

$$\begin{aligned}
\Gamma^{(2)\,\mu\nu\rho\sigma}_h &= S^{(2)\,\mu\nu\rho\sigma}_{\text{EH}}\big|_{\Lambda=0} + S^{(2)\,\mu\nu\rho\sigma}_{\text{gf}} \\
&\quad + \frac{k^2}{32\pi} \left( \Pi^{\text{TL}\,\mu\nu\rho\sigma} \mu_{\text{TL}} - \frac{\Pi^{\text{Tr}\,\mu\nu\rho\sigma}}{(d-1)(d+\gamma_h)^2} \left( A^2 \mu_{\text{TL}} + d(d-2)B^2 \mu_0 \right) \right),
\end{aligned} \tag{21}$$

where $\Pi^{\mathrm{Tr}}$ and $\Pi^{\mathrm{TL}}$ are the trace and traceless projectors, respectively:

$$\Pi^{\mathrm{Tr}}{}_{\mu\nu}{}^{\rho\sigma} = \frac{1}{d}\bar{g}_{\mu\nu}\bar{g}^{\rho\sigma}, \qquad \Pi^{\mathrm{TL}}{}_{\mu\nu}{}^{\rho\sigma} = \delta_{(\mu}{}^{\rho}\delta_{\nu)}{}^{\sigma} - \Pi^{\mathrm{Tr}}{}_{\mu\nu}{}^{\rho\sigma}. \tag{22}$$

The introduction of the dimensionless quantities $\mu_{\mathrm{TL}}$ and $\mu_0$ according to (21) ensures that, in the Landau limit, the two propagating modes feature a standard propagator with mass parameters $\mu_{\mathrm{TL}}$ and $\mu_0$, respectively. We emphasise that Equation (21) is only a rewriting of the two-point function, which conveniently allows us to introduce the individual gaps $\mu_{\mathrm{TL}}$ and $\mu_0$.

To capture the full momentum dependence of the propagators, we introduce independent momentum-dependent wave function renormalisations for the transverse–traceless and the gauge-invariant scalar mode. Specifically, we rescale $h$ according to

$$h_{\mu\nu} \to \mathcal{Z}_{h\,\mu\nu}{}^{\rho\sigma}\, h_{\rho\sigma}, \tag{23}$$

where the wave function renormalisation tensor $\mathcal{Z}_h$, given by

$$\mathcal{Z}_{h\mu\nu}{}^{\rho\sigma} = \delta_{(\mu}{}^{\rho}\delta_{\nu)}{}^{\sigma} + \left(\sqrt{Z_{h^{\mathrm{TT}}}(p^2)} - 1\right)\Pi^{\mathrm{TT}}{}_{\mu\nu}{}^{\rho\sigma} + \left(\sqrt{Z_{h^0}(p^2)} - 1\right)\Pi^0{}_{\mu\nu}{}^{\rho\sigma}. \tag{24}$$

The rescaling (23) entails that the two-point function (21) needs to be multiplied with $\mathcal{Z}_h$ from the left and the right:

$$\Gamma^{(2)}_h \to \mathcal{Z}_h \cdot \Gamma^{(2)}_h \cdot \mathcal{Z}_h. \tag{25}$$

The scale dependence of the graviton wave function renormalisation is encoded in the anomalous dimensions

$$\eta_{\mathrm{TT}}(p^2) = -\partial_t \ln Z_{h^{\mathrm{TT}}}(p^2), \qquad \eta_0(p^2) = -\partial_t \ln Z_{h^0}(p^2). \tag{26}$$

We choose a spectrally adjusted regulator, that is, a regulator that is proportional to the momentum-dependent part of the two-point functions,

$$\mathfrak{R}_k^{h\mu\nu\rho\sigma} = \Gamma_h^{(2)\mu\nu\rho\sigma}\big|_{\mu_{\mathrm{TL}}=\mu_0=0}\mathcal{R}_k(p^2), \tag{27}$$

which ensures that mass-like terms are not regularised [19,35,125,164]. The regulator function $\mathcal{R}_k$ implements the momentum-shell wise integration. Furthermore, we will choose the Landau gauge $\alpha_h \to 0$, which is a fixed point for all gauge parameters [42,165]. With these choices, the graviton propagator reads

$$\begin{aligned}
G_h = {} & \frac{32\pi}{Z_{h^{\mathrm{TT}}}(p^2)}\frac{1}{p^2 + \mathcal{R}_k(p^2) + \mu_{\mathrm{TL}}k^2}\Pi^{\mathrm{TT}} \\
& - \frac{32\pi}{Z_{h^0}(p^2)}\frac{1}{(d-2)(d-1)}\frac{C}{B^2}\frac{1}{p^2 + \mathcal{R}_k(p^2) + \mu_0 k^2}\Pi^0,
\end{aligned} \tag{28}$$

where the coefficients $B$ and $C$ are defined in (19) and (20), respectively.

Similarly, for the ghost, we will distinguish between the longitudinal and the transverse mode, and rescale the corresponding modes with momentum-dependent wave function renormalisations $Z_{c_{\mathrm{L}}}$ and $Z_{c_{\mathrm{T}}}$, respectively. Specifically, the rescaling of the ghost will be

$$c_\mu \to \mathcal{Z}_{c\,\mu}{}^{\alpha} c_\alpha, \qquad \bar{c}_\mu \to \mathcal{Z}_{c\,\mu}{}^{\alpha} \bar{c}_\alpha, \tag{29}$$

with

$$\mathcal{Z}_{c\mu}{}^{\nu} = \sqrt{Z_{c^{\mathrm{T}}}(p^2)}\,\Pi^{\mathrm{T}}{}_{\mu}{}^{\nu} + \sqrt{Z_{c^{\mathrm{L}}}(p^2)}\,\Pi^{\mathrm{L}}{}_{\mu}{}^{\nu}. \tag{30}$$

The longitudinal and transverse projectors are defined in the usual way,

$$\Pi^{\text{L}}{}_{\mu}{}^{\nu} = \frac{p_\mu p^\nu}{p^2}, \qquad \Pi^{\text{T}}{}_{\mu}{}^{\nu} = \delta_\mu{}^\nu - \Pi^{\text{L}}{}_{\mu}{}^{\nu}. \tag{31}$$

The scale dependence of the ghost wave function renormalisations is encoded in the anomalous dimensions

$$\eta_{c^{\text{L}}}(p^2) = -\partial_t \ln Z_{c^{\text{T}}}(p^2), \qquad \eta_{c^{\text{T}}}(p^2) = -\partial_t \ln Z_{c^{\text{L}}}(p^2). \tag{32}$$

Therefore, after distinguishing all different modes on the level of the two-point function and choosing the Landau gauge, there are four momentum-dependent anomalous dimensions and two gaps, completely parameterising the flow of the two-point correlation function.

### 3.2. General Structure of the RG Flows

Since we aim at investigating the flow of physical $n$-point functions, we employ the Landau limit, where the scale dependence of all gauge parameters vanishes. Focusing on the graviton sector in this limit, only the two physical projectors $\Pi^{\text{TT}}$ and $\Pi^0$ contribute to the propagator, see (28). When projecting the external legs of $n$-point functions on the physical modes, the full gauge dependence in the gravity sector is contained in the projector on the scalar mode $\Pi^0$. For this projector, as defined in (17), we observe that

$$\Pi^0[\beta_h, \gamma_h] = \Pi^0 \left[ \frac{\dim \beta_h - \gamma_h}{\dim + \gamma_h}, 0 \right]. \tag{33}$$

This shows that, for the scale dependence of couplings induced by the graviton sector the gauge parameter $\gamma_h$ is redundant, since it can be removed by a rescaling of $\beta_h$.

While the cancellations in the ghost sector are more involved, and require non-trivial cancellations between contributions from propagators and vertices, the same rescaling for $\beta_h$, as given in (33), eliminates the $\gamma_h$-dependence of the ghost-induced flows. Therefore, in the following, we will employ $\gamma_h = 0$ and keep $\beta_h$ general. Despite this redundancy of gauge parameters on the level of the scale dependence of couplings in the Landau limit, the tensor structure corresponding to $\gamma_h$, (6), is nevertheless an independent tensor structure.

With the structure of the graviton propagator as given in (28), together with the regulator defined in (27), we can compute the product of the regulator insertion and two propagators, which enters every diagram of the flow as one of its building blocks, see Figures 1 and 2. For the gravitational sector and in the Landau limit, it reads

$$
\begin{aligned}
G_h \cdot \left( \partial_t \mathfrak{R}_k^h \right) \cdot G_h =\ & \frac{32\pi}{Z_{h^{\text{TT}}}(p^2)} \frac{\partial_t \mathcal{R}_k(p^2) - \eta_{\text{TT}}(p^2)\, \mathcal{R}_k(p^2)}{\left(p^2 + \mathcal{R}_k(p^2) + \mu_{\text{TL}} k^2\right)^2} \Pi^{\text{TT}} \\
& - \frac{32\pi}{Z_{h^0}(p^2)} \frac{1}{(d-2)(d-1)} \frac{C}{B^2} \frac{\partial_t \mathcal{R}_k(p^2) - \eta_0(p^2)\, \mathcal{R}_k(p^2)}{\left(p^2 + \mathcal{R}_k(p^2) + \mu_0 k^2\right)^2} \Pi^0.
\end{aligned}
\tag{34}
$$

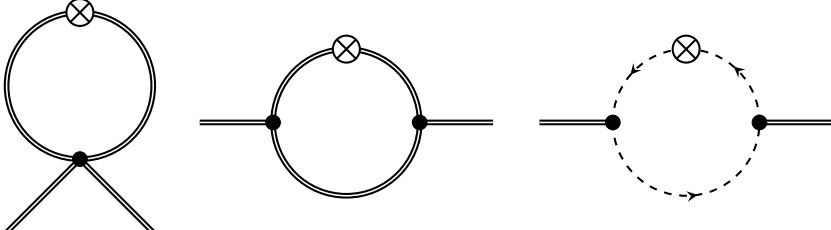

**Figure 1.** Diagrams that contribute to the scale dependence of the graviton two-point function. Double lines indicate gravitons, dashed lines stand for the Faddeev–Popov ghosts, and the circled cross denotes the regulator insertion $\partial_t \mathcal{R}_k$.

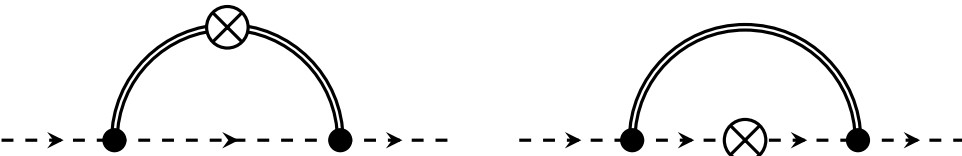

**Figure 2.** Diagrams that contribute to the scale dependence of the ghost two-point function. Double lines indicate gravitons, dashed lines stand for the Faddeev–Popov ghosts, and the circled cross denotes the regulator insertion $\partial_t \mathcal{R}_k$.

A similar expression can be computed for the ghost sector, where the product can be spanned by a transverse and a longitudinal part. Importantly, in the Landau limit, both the graviton propagator and the above product decay into a sum of the two projectors $\Pi^{\text{TT}}$ and $\Pi^0$. This feature is due to a cancellation of gauge modes contained in the regulator with contributions from the propagators. It entails that the flow of $n$-point functions projected on gauge-invariant modes is only driven by the gauge-invariant modes themselves, and no gauge modes drive their scale dependence. It also significantly decreases the number of different tensor structures that need to be computed, when generalising the present computation to higher $n$-point correlators. Due to the decomposition into the orthogonal projectors $\Pi^{\text{TT}}$ and $\Pi^0$, vertices with one or more gauge mode will not contribute to the scale dependence of any physical $n$-point correlator [107].

In the graviton sector, we project onto the different tensor structures of the two-point function by contracting the indices of (21) with $\Pi^{\text{TT}}$ and $\Pi^0$, respectively. Structurally, after projection, the flow of the TT-part of the graviton two-point function reads [29,52]:

$$
- (y + \mu_{\text{TL}})\eta_{\text{TT}}(y) + \partial_t \mu_{\text{TL}} + 2\mu_{\text{TL}} = \frac{1}{k^2 Z_{h^{\text{TT}}}(p^2)} \Pi^{\text{TT}}{}_{\mu\nu}{}^{\rho\sigma} \dot{\Gamma}^{(2)}_{h}{}_{\rho\sigma}{}^{\mu\nu} \equiv \text{flow}_{\text{TT}}(y), \quad (35)
$$

where the last term on the left-hand side comes from the scale derivative acting on the $k^2$ coming with the dimensionless gap $\mu_{\text{TL}}$, see (21). Furthermore, the right-hand side is obtained by evaluating and projecting the diagrams in Figure 1. We also introduced the shorthand $y = \frac{p^2}{k^2}$.

In a complete analogy, the flow equation for the scalar mode of the graviton two-point function reads

$$
- (y + \mu_0)\eta_0(y) + \partial_t \mu_0 + 2\mu_0 = \frac{1}{k^2 Z_{h^0}(p^2)} \Pi^0{}_{\mu\nu}{}^{\rho\sigma} \dot{\Gamma}^{(2)}_{h}{}_{\rho\sigma}{}^{\mu\nu} \equiv \text{flow}_0(y). \quad (36)
$$

Each of the Equations (35) and (36) is disentangled by evaluating the right-hand-side at $y = -\mu_{\text{TL}}$ and $y = -\mu_0$, respectively [29]:

$$
\partial_t \mu_{\text{TL}} = -2\mu_{\text{TL}} + \text{flow}_{\text{TT}}(-\mu_{\text{TL}}), \quad \partial_t \mu_0 = -2\mu_0 + \text{flow}_0(-\mu_0), \quad (37)
$$

and

$$
\eta_{\text{TT}}(y) = -\frac{\text{flow}_{\text{TT}}(y) - \text{flow}_{\text{TT}}(-\mu_{\text{TL}})}{y + \mu_{\text{TL}}}, \quad \eta_0(y) = -\frac{\text{flow}_0(y) - \text{flow}_0(-\mu_0)}{y + \mu_0}. \quad (38)
$$

In the ghost sector, the equations determining the anomalous dimensions are

$$
y\,\eta_{c^{\text{T}}}(y) = \frac{1}{k^2 Z_{c^{\text{T}}}(p^2)} \Pi^{\text{T}}{}_{\mu}{}^{\nu} \dot{\Gamma}^{(2)}_{c}{}_{\nu}{}^{\mu} \equiv \text{flow}_{c^{\text{T}}}(y),
$$
$$
y\,\eta_{c^{\text{L}}}(y) = \frac{1}{k^2 Z_{c^{\text{L}}}(p^2)} \Pi^{\text{L}}{}_{\mu}{}^{\nu} \dot{\Gamma}^{(2)}_{c}{}_{\nu}{}^{\mu} \equiv \text{flow}_{c^{\text{L}}}(y),
$$
$$
\quad (39)
$$

where the diagrammatic representation of flow$_c(y)$ is shown in Figure 2. The transverse and longitudinal parts of the flow are extracted by contraction with the corresponding projectors (31).

To evaluate the diagrams shown in Figures 1 and 2, we used the Mathematica packages *xAct* [166–170]. The complete derivation is included in the attached notebook. We validated the correctness of the results with an independent code based on *xAct* as well as *FormTracer* [171].

### 3.3. On the Relation between Form Factors and Anomalous Dimensions

We will now briefly discuss the relationship between an action containing form factors and the momentum-dependent wave function renormalisations. On a flat background, the complete information on the graviton propagator is included in the action

$$\Gamma \simeq \frac{1}{16\pi G_N} \int d^d x \sqrt{g} \left[ 2\Lambda - R - \frac{1}{4} \frac{d-2}{d-1} R f_R(\Delta) R + \frac{1}{4} \frac{d-2}{d-3} C^{\mu\nu\rho\sigma} f_C(\Delta) C_{\mu\nu\rho\sigma} \right], \quad (40)$$

where $f_R$ and $f_C$ are form factors. The normalisation of the form factors ensures a unit prefactor in the propagator, see (41) below. This action, which is based on a curvature expansion of the full effective action, is an expansion in terms of diffeomorphism invariant operators. The form factors $f_R$ and $f_C$ capture the full momentum dependence of the graviton propagator. In comparison to a derivative expansion, they do not necessarily introduce new poles into the propagator. In a derivative expansion, one would in general expect the emergence of additional, potentially spurious poles of the propagators (see [120], and Section 5.5). However, computations based on curvature expansions usually make use of the background-field approximation, where the difference between the background and the fluctuation propagators are neglected. This approximation potentially suffers from severe background dependence, which is lifted in the fluctuation expansion of the effective action (10).

Since the gauge fixing and the regulator both break diffeomorphism invariance, the expansion in terms of form factors, as in (40) and in terms of fluctuation vertex correlation functions, do not agree. Therefore, the question arises whether it is possible to extract the diffeomorphism invariant part—the form factors—from the scale dependences of the fluctuation two-point functions, without explicitly computing and solving modified Ward identities.

In the following, we will present a mapping between the form factors $f_R$ and $f_C$, and the fluctuation two-point functions. For this, we will compare the propagators in both expansions, and neglect that these propagators are different due to the breaking of diffeomorphism invariance. A motivation for this assumption is the feature of effective universality [64], which suggests that, at least in the vicinity of the fixed point solution, the breaking of diffeomorphism invariance is mild. Therefore, under this assumption, we investigate the possibility of extracting the form factors $f_R$ and $f_C$ from the fluctuation propagators.

For this, we will briefly indicate the form of the (unregularised) flat background propagator that arises from the action (40). We will also set the cosmological constant to zero for the moment. In this case, the scalar parts of the spin two and zero background propagators read

$$\bar{G}^{\rm TT}(p^2) \propto \frac{1}{p^2(1 + p^2 f_C(p^2))}, \qquad \bar{G}^0(p^2) \propto \frac{1}{p^2(1 + p^2 f_R(p^2))}. \quad (41)$$

This can be compared with the unregularised fluctuation propagator (28) at vanishing gaps, and suggests the following relation between form factors and wave function renormalisation:

$$\frac{Z_{h^{\rm TT}}(p^2)}{Z_{h^{\rm TT}}(0)} = 1 + p^2 f_C(p^2), \qquad \frac{Z_{h^0}(p^2)}{Z_{h^0}(0)} = 1 + p^2 f_R(p^2). \quad (42)$$

For finite values of the cosmological constant and the gaps, the situation is more complicated. The cosmological constant enters the two parts of the background propagators in a specific (gauge-dependent) ratio. By contrast, in general, the fluctuation gaps need to not have any particular ratio. It is thus, in general, not possible to map the propagators in a one-to-one way without non-local terms in at least one of the form factors. If we nevertheless want to insist on an equivalence and allow for inverse powers of the momentum in at least one of the form factors, we can still map the propagators one-to-one. For example, we could identify the gap in the spin two sector with the actual cosmological constant, and introduce a term $\sim 1/p^4$ in the form factor $f_R$. Interestingly, such a term has been observed in the context of reconstructing an effective action from numerical lattice data obtained within causal dynamical triangulations [172]. Such a term also has interesting cosmological applications [173,174]. We will, however, not further discuss this issue here, since in the end, a proper evaluation of the problem necessarily involves the solution of the modified Ward identities.

### 3.4. Momentum-Dependent Anomalous Dimension versus Wave Function Renormalisation

As a final point in this section, we will discuss the relation of the fixed point condition for the dimensionless wave function renormalisation and the momentum-dependent anomalous dimensions. In the literature on momentum dependence in asymptotic safety, one commonly uses momentum-dependent anomalous dimensions rather than the wave function renormalisation. This is because only the anomalous dimensions are relevant in the flow equations—by constructing all factors of the wave function renormalisation drop out, see for example, (35). Since the anomalous dimension is related to the scale derivative of the dimensionful wave function renormalisation, one does not impose the fixed point condition directly. We can, however, easily translate between this language and a formulation, where we require the dimensionless wave function renormalisation to be constant at the fixed point with respect to the RG scale $k$, see also [66].

The definition of the anomalous dimension $\eta$ in terms of the dimensionful wave function renormalisation $Z$ reads

$$\eta(y) = -\partial_t \ln Z(y)\,, \tag{43}$$

see (26) and (32). We want to express the right-hand side in terms of the dimensionless wave function renormalisation $z$, for which we require a fixed point, and which is given by [1]

$$z(y) = k^{\eta(0)} Z(y)\,. \tag{44}$$

In that process, we have to account both for the fact that we can normalise the wave function renormalisation—giving rise to an anomalous running induced by the anomalous dimension evaluated at zero momentum—as well as the scaling of the argument. In that way, we find

$$\eta(y) = \eta(0) - \frac{\dot{z}(y)}{z(y)} + 2y\frac{z'(y)}{z(y)}\,. \tag{45}$$

Here, the overdot indicates the logarithmic scale derivative with respect to the intrinsic $k$-dependence, excluding the trivial scaling of the argument. At a fixed point, where $\dot{z}_* = 0$, we have

$$\eta_*(y) = \eta_*(0) + 2y\frac{z'_*(y)}{z_*(y)}\,. \tag{46}$$

We can solve this differential equation for the wave function renormalisation:

$$z_*(y) = z_*(0)e^{\int_0^y \mathrm{d}s \frac{\eta_*(s)-\eta_*(0)}{2s}} = z_*(0)e^{\int_0^1 \mathrm{d}\omega \frac{\eta_*(\omega y)-\eta_*(0)}{2\omega}}\,. \tag{47}$$

Assuming that the momentum-dependent anomalous dimension is bounded, this has the consequence that for large $y$, the wave function renormalisation scales as

$$z_*(y) \propto y^{\frac{\eta_*(\infty)-\eta_*(0)}{2}}, \qquad \text{as } y \to \infty.$$ (48)

Notably, only if the anomalous dimension vanishes asymptotically, that is, if it fulfils *momentum locality*[2] [31], the standard fall-off behaviour of the propagator in terms of the anomalous dimension at zero follows, see (28), namely

$$G(y) \propto \frac{1}{y^{1-\frac{\eta(0)}{2}}}, \qquad \text{as } y \to \infty. \qquad \text{(momentum locality)}$$ (49)

If momentum locality is not fulfilled, we need non-local information on the momentum dependence, and the formula reads

$$G(y) \propto \frac{1}{y^{1+\frac{\eta(\infty)-\eta(0)}{2}}}, \qquad \text{as } y \to \infty. \qquad \text{(no momentum locality)}$$ (50)

Having direct access to the fixed point wave function renormalisation via (47), we can calculate the fixed point form factor from the corresponding momentum-dependent anomalous dimension by inverting (42),

$$f_*(y) = \frac{e^{\int_0^y \mathrm{d}s\, \frac{\eta_*(s)-\eta_*(0)}{2s}} - 1}{y}.$$ (51)

Consequently, the behaviour at the large momentum is

$$f_*(y) \sim \begin{cases} cy^{\frac{\eta_*(\infty)-\eta_*(0)}{2}-1}, & \eta_*(\infty) - \eta_*(0) > 0, \\ -\frac{1}{y} + cy^{\frac{\eta_*(\infty)-\eta_*(0)}{2}-1}, & \eta_*(\infty) - \eta_*(0) \le 0, \end{cases} \qquad \text{as } y \to \infty,$$ (52)

where $c$ is given by

$$c = e^{\int_0^\infty \mathrm{d}s \left[ \frac{\eta_*(s)-\eta_*(0)}{2s} - \frac{\eta_*(\infty)-\eta_*(0)}{2(1+s)} \right]}.$$ (53)

For a derivation of this, see Appendix A.

Let us finally make the connection to the derivative expansion, see also [29]. To quadratic order in $y$, we find

$$\frac{z_*(y)}{z_*(0)} = 1 + \frac{1}{2}\eta'_*(0)\, y + \frac{1}{8}\left( \eta'_*(0)^2 + \eta''_*(0) \right) y^2 + \mathcal{O}(y^3).$$ (54)

This translates into the form factor

$$f_*(y) = \frac{1}{2}\eta'_*(0) + \frac{1}{8}\left( \eta'_*(0)^2 + \eta''_*(0) \right) y + \mathcal{O}(y^2).$$ (55)

Let us also emphasise at this point that, in theories with more than one mode and correspondingly more than one anomalous dimension, the derivative expansion is potentially unstable—the more so the more modes one has [175]. This comes from the fact that, for a well-defined flow, one needs that for all modes $z_*(y) > 0$. In particular, this implies that all the highest retained coefficients of all form factors need to have a positive sign to avoid turning one of the modes into a ghost. For general theories, there is, however, no reason why all these coefficients need to be positive. One therefore likely faces the dilemma that, for a given order in the derivative expansion, the highest order coefficient of some mode might be negative. With a large number of modes, it is thus more likely that at no finite order, a fixed point, which is physically viable and does not necessarily feature ghost modes, can be found in a derivative expansion. This makes it clear that in complicated

systems, the full resolution of the momentum dependence is not optional. In fact, as we will see in Section 5, we find indications that quantum gravity with its two off-shell modes already shows such an alternating pattern for low orders of the derivative expansion.

## 4. Momentum-Dependent Fluctuation RG Flow: Analytical Structure

Now, we will discuss several analytical properties of the momentum-dependent fluctuation two-point correlation functions which were introduced in Section 3. For a clearer notation, we will work with dimensionless momenta, that is, we make the identification $\frac{p^2}{k^2} \to p^2$, such that no explicit factors of $k$ will appear in the following discussion.

As has been found earlier, the Landau limit $\alpha_h \to 0$ induces a fixed point for all gauge parameters [42]. Due to this fact, we only consider this limit for the fluctuation flows in this work. Additionally, there is the constraint

$$\beta_h < d - 1 \, , \tag{56}$$

which ensures that the ghost kinetic operator has the correct sign and is invertible. The most popular gauge choice in the literature is

$$\beta_h = \frac{d}{2} - 1 \, , \tag{57}$$

which we will focus on in our analysis. Other choices that have been argued for to be preferred are the transverse gauge $\beta_h = -1$ [154,176] and the singular choice $\beta_h \to -\infty$, which has been considered in, for example, [35,42,138,139]. The choice $\beta_h = 0$ in the Landau limit allows for a particularly simple regularisation in a curved spacetime [48].

The analytical properties, which we will discuss in the following, will provide important cross-checks to test the numerical evaluation discussed in Section 5.

### 4.1. Behaviour at Small Momentum

Let us first discuss the behaviour of the flow of the two-point function at small momentum. A key observation is that the flow contains terms of the form

$$\frac{1}{(p^2 + 2pqx + q^2)} \, , \qquad \frac{1}{(p^2 + 2pqx + q^2)^2} \, , \tag{58}$$

where $p$ is the external momentum, $q$ is the loop momentum, and $x$ is the cosine of the angle between the two momenta. These terms look like unregulated, gapless propagators. Their origin lies in the projectors occurring in the propagators, see (16) and (17), and are thus simply a feature of gravity. The same phenomenon happens for spin one flows, where the transverse and longitudinal projectors also introduce such terms. These terms cause a number of technical difficulties, see also the discussion below in Section 4.4. For small external momenta $p$, they are the reason that the derivative expansion does not commute with performing the loop integral. This is because expanding these terms in small $p$ is actually an expansion in powers of $p^2/q^2$, so that higher orders of the external momentum introduce higher negative powers of the loop momentum. At a critical order, the negative powers cancel the measure term $q^{d-1}$ and potential extra powers from the vertex factors. For higher orders, the integrals thus do not converge as they suffer from IR divergences. This problem has been encountered before [177], where momentum-independent contributions of vertices have been neglected to define a finite, but ultimately inconsistent, flow. Even at a level where the integrals converge, strong instabilities have been found [29] which might prevent one from obtaining conclusive and robust results.

Note that once the integral over the loop momentum has been performed, the flow appears to be smooth (see the numerical results presented below in Section 5). This emphasises that reliable approximations must involve momentum dependence beyond a local expansion.

Besides this general discussion, one can investigate whether there are relations between the different flows at vanishing momentum. In the graviton sector, we find the relation

$$\beta_h \to -\infty: \qquad \left.\frac{\text{flow}_{\text{TT}}}{\text{flow}_0}\right|_{p^2=0} = \frac{2}{d} - 1 < 0. \tag{59}$$

This limit is formally singular, in part due to how we have to introduce the gap in the spin zero sector, see (21). In this limit, our ansatz formally diverges, indicating that we actually write down a gap for the spin zero gauge mode. This is consistent as this gauge choice projects out the trace mode.

In the ghost sector, we find that

$$\left.\frac{\text{flow}_{c^{\text{T}}}}{\text{flow}_{c^{\text{L}}}}\right|_{p^2=0} = \frac{d-1-\beta_h}{d-1} > 0. \tag{60}$$

The divergence for $\beta_h \to -\infty$ indicates that the longitudinal flow vanishes at $p = 0$ for this gauge choice. Due to the constraint (56), the flows, and thus the ghost anomalous dimensions, have the same sign at vanishing momentum.

*4.2. Behaviour at Large Momentum*

We will now discuss the large momentum behaviour of the flow of the two-point function. We emphasise that all of the following aspects a priori rely on our truncation of the three- and four-point function—a dynamical implementation of their flow can alter these results. In the following, we also assume that the regulator falls off exponentially. Similar conclusions hold for regulators with compact support, in which case no exponentially suppressed corrections appear. Some of the aspects that we present in the following have been previously discussed in [107,178].

As a general feature, we note that in the large momentum limit, the self-energy diagrams simplify. This is because of our assumption on the regulator, so that we can neglect it in the propagator which carries both the loop and the external momentum, up to exponentially small corrections. As a consequence, we can perform the angular integration exactly. The relevant integrals are

$$\int_{-1}^{1} \mathrm{d}x \left(1 - x^2\right)^{\frac{d-3}{2}} \left(p^2 + 2pqx + q^2\right)^k \left(p^2 + 2pqx + q^2 + \mu\right)^{-1}, \qquad k \in \{-2, \dots, 6\}, \tag{61}$$

which give rise to hypergeometric functions. We will now discuss the different parts of the flow in this limit in turn.

4.2.1. Spin Two Sector

It has been noted before that in $d = 4$ and with identified three- and four-graviton couplings, the spin two two-point function shows momentum locality [29,31,39]. This means that the flow of the two-point function goes to a constant at large momentum, in contrast to the naive expectation of a quadratic behaviour. This is due to a cancellation of the self-energy and the tadpole diagrams.

We found that this cancellation only happens in $d = 4$—in higher dimensions, the two-point function rises quadratically at large momentum. We can also confirm that this qualitative behaviour is independent of the choice of gauge parameters and regulators. When we discuss the numerical results below, this manifests itself by $\eta_{\text{TT}}$ going to zero for large arguments in $d = 4$, but staying finite in this limit in other dimensions, see Figure 3.

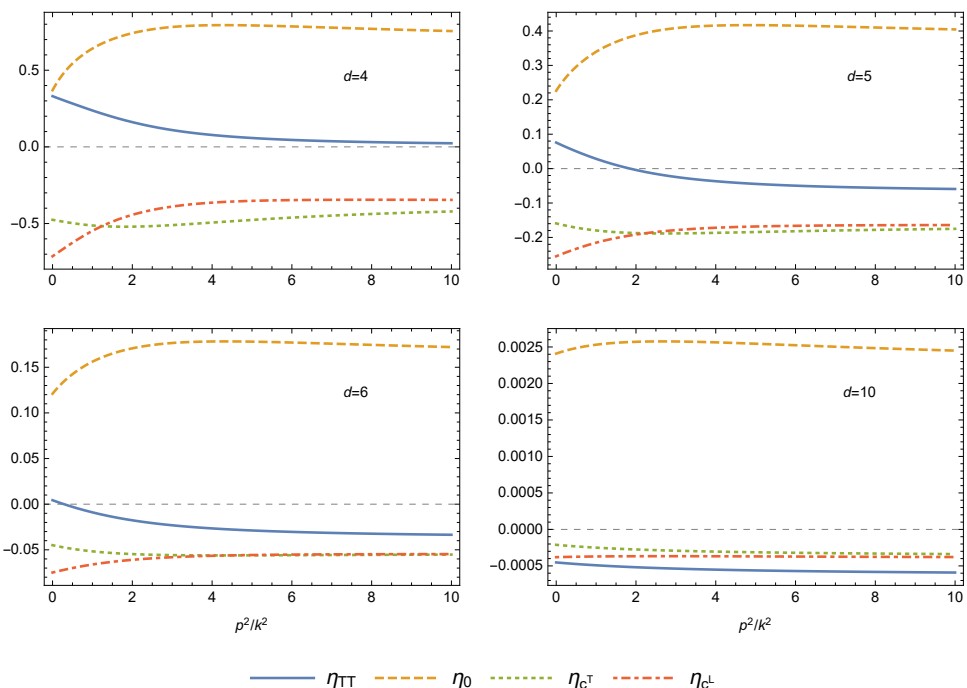

**Figure 3.** Dimensional dependence of anomalous dimensions for the choices $g = 1$, $\mu_{\mathrm{TL}} = 0$, $\mu_0 = 0$, $\beta_h = \frac{d}{2} - 1$ and the regulator (83). For clarity, we have reinstated $k$ in the momentum argument.

There are three contributions to the flow of the spin two two-point function: the graviton tadpole, the graviton self-energy, and the ghost self-energy diagram, see Figure 1. The tadpole diagram has the exact form

$$\mathrm{flow}_{\mathrm{TT}}^{\mathrm{tadpole}}(p^2, \mu_{\mathrm{TL}}, \mu_0) = g_4 \left( A_{\mathrm{TT},0}^{\mathrm{tadpole}}[\eta_{\mathrm{TT}}, \eta_0](\mu_{\mathrm{TL}}, \mu_0) + A_{\mathrm{TT},2}^{\mathrm{tadpole}}[\eta_{\mathrm{TT}}, \eta_0](\mu_{\mathrm{TL}}, \mu_0)p^2 \right). \tag{62}$$

The coefficients $A_{\mathrm{TT},i}^{\mathrm{tadpole}}$ depend functionally on the graviton anomalous dimensions and the regulators, as well as on the gaps, gauge parameters and the dimension. The structure of the self-energy diagrams is more involved due to the propagator depending on the sum of external and loop momentum. In the large external momentum limit, neglecting the regulator depending on the sum of the two momenta, we find that

$$\mathrm{flow}_{\mathrm{TT}}^{\mathrm{hSE}}(p^2, \mu_{\mathrm{TL}}, \mu_0) \sim g_3 \int \frac{\mathrm{d}^d q}{(2\pi)^d} \sum_{i=-2}^{6} \left[ A_{\mathrm{TT},i}^{\mathrm{hSE}}[\eta_0](p^2, q^2) \frac{(p^2 + 2pqx + q^2)^i}{p^2 + 2pqx + q^2 + \mu_0} \right. \tag{63}$$
$$\left. + B_{\mathrm{TT},i}^{\mathrm{hSE}}[\eta_{\mathrm{TT}}](p^2, q^2) \frac{(p^2 + 2pqx + q^2)^i}{p^2 + 2pqx + q^2 + \mu_{\mathrm{TL}}} \right], \qquad \text{as } p \to \infty,$$

for the graviton self-energy diagram, and

$$\mathrm{flow}_{\mathrm{TT}}^{\mathrm{cSE}}(p^2) \sim g_c \int \frac{\mathrm{d}^d q}{(2\pi)^d} \sum_{i=-2}^{4} A_{\mathrm{TT},i}^{\mathrm{cSE}}[\eta_{\mathrm{c}^{\mathrm{T}}}, \eta_{\mathrm{c}^{\mathrm{L}}}](p^2, q^2)\left(p^2 + 2pqx + q^2\right)^i, \qquad \text{as } p \to \infty. \tag{64}$$

for the ghost self-energy diagram. Performing the integrals, we find the behaviour

$$\mathrm{flow}_{\mathrm{TT}}(p^2, \mu_{\mathrm{TL}}, \mu_0) \sim p^2 \left[ (d-4)g_3 \mathcal{I}_{\mathrm{TT}}^1[\eta_{\mathrm{TT}}, \eta_0](\mu_{\mathrm{TL}}, \mu_0) \right.$$
$$\left. + (g_4 - g_3)\mathcal{I}_{\mathrm{TT}}^2[\eta_{\mathrm{TT}}, \eta_0](\mu_{\mathrm{TL}}, \mu_0) \right], \text{ as } p \to \infty. \tag{65}$$

The $\mathcal{I}_{\mathrm{TT}}^i$ are gauge- and dimension-dependent functions. This parameterisation exemplifies that, only for $g_3 = g_4$, is the flow momentum-local in $d = 4$. The ghost diagram does

not contribute to the leading order behaviour. We highlight that the agreement of $g_3$ and $g_4$ at the asymptotically fixed point was indeed discovered on a semi-quantitative basis in [39]. This semi-quantitative agreement between different $n$-point correlators, also discovered in the context of gravity-matter systems [64–66], might indicate the near-perturbative nature of asymptotically safe quantum gravity.

More generally, one can ask if there are other (integer) dimensions where there is a choice $g_4 = cg_3$ for which the flow exhibits momentum locality. This is indeed the case for

$$g_4 = \frac{7}{4}g_3, \qquad d = 6. \tag{66}$$

This is the only other combination of coupling identifications and integer dimensions, which gives rise to momentum locality.

### 4.2.2. Spin Zero Sector

The general structure of the spin zero sector is the same as that of the spin two sector. In particular, asymptotically we have an expansion as in (62)–(64) for the different diagrams. The main difference is that we do not find momentum locality in $d = 4$. The only combination of coupling identification and integer dimension, which gives rise to a momentum-local spin zero sector is

$$g_4 = -2g_3, \qquad d = 6. \tag{67}$$

From this we conclude that, at least in our setup, there is no situation where both spin two and spin zero sector are momentum-local. However, if higher $n$-point functions are also distinguished according to their tensor structures, this situation might change.

One can further ask the question, for $d \neq 4$, whether there is a specific ratio between the behaviour of the flows of the two sectors at large momentum. For this analysis, we assume again that $g_4 = g_3$. In general, there is no such relation: the contributions, including the spin zero propagator to each of the flows is generally different from the contributions including the spin two propagator. There are three exceptions to this. Two of them are independent of the choice of the gauge parameter $\beta_h$:

$$
\begin{aligned}
d = 3: && \frac{\text{flow}_{\text{TT}}}{\text{flow}_0}(p^2, \mu_{\text{TL}}, \mu_0) &\sim 1, && \text{as } p \to \infty, \\
d = 6: && \frac{\text{flow}_{\text{TT}}}{\text{flow}_0}(p^2, \mu_{\text{TL}}, \mu_0) &\sim -\frac{1}{4}, && \text{as } p \to \infty.
\end{aligned}
\tag{68}
$$

The third exception is the gauge choice $\beta_h \to -\infty$, so that

$$\beta_h \to -\infty: \quad \frac{\text{flow}_{\text{TT}}}{\text{flow}_0}(p^2, \mu_{\text{TL}}, \mu_0) \sim -\frac{(d-4)(d^3 - d^2 + 8d - 12)}{(d-2)(d+2)(3d^2 - 11d + 12)}, \quad \text{as } p \to \infty. \tag{69}$$

### 4.2.3. Ghost Sector

The flow in the ghost sector differs structurally from the flow in the graviton sector in that the tadpole diagram vanishes for a linear parameterisation of the metric fluctuations [18], see also (8), since in the present approximation the ghost action is linear in $h$. Neither of the two ghost modes shows momentum locality in our setup for any choice of dimension and gauge. However, independent of dimension and gauge choice, the two flows agree asymptotically,

$$\frac{\text{flow}_{c^{\text{T}}}}{\text{flow}_{c^{\text{L}}}}(p^2, \mu_{\text{TL}}, \mu_0) \sim 1, \qquad \text{as } p \to \infty. \tag{70}$$

In this limit, only one of the two self-energy diagrams in Figure 2 contributes, namely the first one, where the regulator is inserted on the graviton line. Together with the relation of the two flows at vanishing momentum, this gives an estimate of their typical disagreement.

*4.3. Limit of Large Dimension*

Let us discuss aspects of the limit $d \to \infty$. In this limit, we can perform the remaining angular integral exactly, and also comment on the radial integral. To see the first statement, note that this integral structurally looks like

$$\int_{-1}^{1} dx \left(1 - x^2\right)^{\frac{d-3}{2}} f(x) ,\tag{71}$$

where $f(x)$ is one of the integrands in the flow. We will now make the assumption that the dependence on $x$ is smooth, so that $f$ admits a convergent expansion in Chebyshev polynomials,

$$f(x) = \sum_{n \geq 0} f_n T_n(x) .\tag{72}$$

Inserting this expansion into (71), and assuming that we can swap the order of summation and integration, we arrive at

$$\int_{-1}^{1} dx \left(1 - x^2\right)^{\frac{d-3}{2}} f(x) = \sum_{n \geq 0} f_n \int_{-1}^{1} dx \left(1 - x^2\right)^{\frac{d-3}{2}} T_n(x)$$

$$= \sum_{k \geq 0} f_{2k} \left(-\frac{1}{2}\right)^k \frac{\sqrt{\pi}\, \Gamma\left(\frac{d-1}{2}\right)}{\Gamma\left(\frac{d}{2} + k\right)} \frac{(d-2)!!}{(d-2k-2)!!}\tag{73}$$

$$\sim \sqrt{\frac{2\pi}{d}} \sum_{k \geq 0} (-1)^k f_{2k} = \sqrt{\frac{2\pi}{d}} f(0) , \qquad \text{as } d \to \infty .$$

In the last line, we expanded the expression to leading order in the limit of large $d$. This result is intuitively clear: for large $d$ the measure suppresses all values of $x$ except $x = 0$.

To deal with the radial integral, we will assume that we have a regulator which falls off like an exponential, so that also all integrands in the flow fall off exponentially. More precisely, with $z$ denoting the square of the loop momentum, we assume that any integrand is of the form

$$f(z) = g(z)(1 + z)^k e^{-az} , \qquad a > 0, \, k \geq 0 .\tag{74}$$

We included an additional power law behaviour so that we can assume that $g$ is bounded on the whole interval. As a consequence, we can expand it in a series of rational Chebyshev functions,

$$g(z) = \sum_{n \geq 0} g_n T_n \left(\frac{z-1}{z+1}\right) .\tag{75}$$

Combining this with the measure, we have integrals of the form

$$\sum_{n \geq 0} g_n \int_0^{\infty} dz^{\frac{d}{2}+k} T_n \left(\frac{z-1}{z+1}\right) e^{-az} \sim \sum_{n \geq 0} g_n a^{-\frac{d}{2}-1-k} 2\sqrt{\pi} \left(\frac{d}{2}\right)^{\frac{d}{2}+1+k} e^{-\frac{d}{2}}$$

$$= 2\sqrt{\pi} g(\infty) e^{-\frac{d}{2}} \left(\frac{d}{2a}\right)^{\frac{d}{2}+1+k} , \qquad \text{as } d \to \infty .\tag{76}$$

If we combine these two results with the factor from the angular integration,

$$\int d\Omega = \frac{1}{2^d \pi^{\frac{d+1}{2}} \Gamma\left(\frac{d-1}{2}\right)} ,\tag{77}$$

we get the overall scaling of the flow as

$$\text{flow} \propto \left( \frac{1}{4\pi a} \right)^{\frac{d}{2}} d^{k+1} . \tag{78}$$

This result shows that there are two outcomes for the behaviour at large $d$, depending on the fall-off of the regulator. For

$$a > \frac{1}{4\pi} , \tag{79}$$

the dimensional factor goes to zero, whereas for

$$a \leq \frac{1}{4\pi} , \tag{80}$$

the flow diverges in this limit.

*4.4. The Flow for Positive $\mu$*

As a last point in this section, we will discuss a technical challenge related to the projection procedure for the anomalous dimensions, see (35) and (36). For positive gaps $\mu > 0$, we have to evaluate the flow at negative squared momentum. This is, in general, challenging, for two reasons. First, we have to define a regulator for, in general, complex momenta. Second, the factors (58) induce poles in the integration domain for $p^2 < 0$.

We choose a regulator in the following way. First, to avoid having to define it in the entire complex plane, we take the real part of the argument. Second, we have to define the desired behaviour of the regularised propagator for $p^2 = -\mu$, that is, we have to define

$$\left. \frac{1}{p^2 + \mathcal{R}_k(p^2) + \mu} \right|_{p^2 = -\mu} = \frac{1}{\mathcal{R}_k(-\mu)} . \tag{81}$$

In the limit of large masses, we require that this expression behaves like a standard regularised propagator,

$$\mathcal{R}_k(-\mu) \sim 1 + \mu , \qquad \text{as } \mu \to \infty . \tag{82}$$

Third, we clearly also need that the regularised propagator does not introduce any poles in the integration region. A choice that fulfils all these requirements is

$$\mathcal{R}_k(y) = \frac{e^{-\tilde{y}}}{1 + e^{-2\tilde{y}}} + \frac{1 - \tilde{y}}{1 + e^{2\tilde{y}}} , \qquad \tilde{y} = \text{Re } y . \tag{83}$$

For (large) positive arguments, this regulator is just of an exponential type since the second term vanishes in this limit.

Let us now discuss how we deal with the unregularised pole structures (58) coming from the projectors. The poles are situated at $x = 0$ and $q^2 = -p^2 = \mu$. We define the integration over these poles by splitting the integration domain into a disk centred around the pole, and the rest. Inside the disk, we choose radial coordinates. One can show analytically that the Jacobian of the coordinate transformation and the angular integration remove all potential poles, so that the integral is well-defined. Let us finally mention that only simple poles arise in approximations that do not resolve the difference between the two graviton anomalous dimensions and gaps.

## 5. Momentum-Dependent Fluctuation RG Flow: Numerical Results

In this section, we discuss numerical results on the momentum dependence of the propagators, and the influence of the dimension, the choice of gauge and the size and sign of gaps. We will also discuss the derivative expansion of the form factors. All results are

obtained with the regulator (83). As a generic choice, we set the gravitational coupling to one, $g = 1$, and use the Landau limit $\alpha_h \to 0$.

### 5.1. Numerical Strategy

Before we present the actual numerical results, we provide a short discussion of how we obtained them. In the previous section, we found that the anomalous dimensions are bounded both at zero and infinite arguments. We will further assume that they are bounded on the entire positive real line. In that case, they can be expanded in a series of rational Chebyshev functions [179],

$$\eta(y) = \sum_{n \geq 0} \eta_n T_n \left( \frac{y - 1}{y + 1} \right). \tag{84}$$

Such an expansion is equivalent to compactifying the domain and employing a standard expansion in Chebyshev polynomials. This expansion shows desirable convergence properties if the function that is represented by the series is smooth, see for example, [180] for an in-detail discussion. In the context of functional renormalisation group flows, they have been systematically discussed in [181–183]. For applications in quantum gravity, see also [68,101,145].

In practice, we will truncate (84) at a finite order, insert the expansion into the integral equations, and evaluate the equations at a set of collocation points. The set of integral equations for the anomalous dimensions then reduces to an *algebraic* set of equations for the expansion coefficients, which can be solved by standard linear algebra methods. The integrals have been performed with Mathematica's numerical integration routine.

To judge whether the numerical precision of the solution is high enough, we verify that the analytic relations between the different anomalous dimensions found in the last section are satisfied.

Note that the above approach is non-perturbative in the coupling $g$, and works for any value of it. If one is only interested in the anomalous dimensions for small $g$, one can perform a Taylor series in $g$, the coefficients of which are nested integrals that can be computed numerically. In that way, the overall magnitude of the anomalous dimensions is directly controlled by $g$, at least for small enough values.

### 5.2. Dimensional Dependence

First, we will study the dimensional dependence of the anomalous dimensions. For the gauge choice $\beta_h = \frac{d}{2} - 1$ and vanishing gaps $\mu_{\mathrm{TL}} = \mu_0 = 0$; this is shown in Figure 3. There are a few general features. First, the overall magnitude of the anomalous dimensions decreases with increasing dimension. This is in agreement with our analytical estimate in the previous section, see (78). Second, we observe that while $\eta_0$ stays positive at these coordinates in all dimensions, $\eta_{\mathrm{TT}}$ shifts to negative values quickly. Third, while there is some quantitative difference between the two ghost anomalous dimensions in $d = 4$, they agree more and more the larger the dimension is. Fourth, as we have seen analytically in Section 4.2, $\eta_{\mathrm{TT}}$ is momentum-local only in $d = 4$. Finally, we observe a general flattening for large $d$: the momentum dependence of all anomalous dimensions is essentially trivial so that they can be very well approximated by a constant. However, this conclusion only holds in the current approximation, where higher-order curvature operators were neglected in the seed action. Since some of these operators are expected to be relevant in higher dimensions, their presence might alter the momentum dependence of the anomalous dimensions.

### 5.3. Gap Dependence

Let us now discuss the dependence of the anomalous dimensions on the gaps $\mu_{\mathrm{TL}}$ and $\mu_0$. For this, we fix $d = 4$ and the gauge parameters $\beta_h = 1$. The anomalous dimensions are shown in Figure 4 for the gaps $\mu_x = \pm 1/4$.

Generally, we see that the gaps mostly influence the overall magnitude and the behaviour at small momenta. Rather small variations in the gaps can shift the value of

$\eta_0(0)$ rather drastically. By contrast, the overall shape of the other anomalous dimensions is rather unaffected. We take this as evidence that the qualitative momentum dependence in large parts of theory space looks approximately similar to that at the specific points that we present in this work.

### 5.4. Gauge Dependence

Now we will discuss the gauge dependence of the anomalous dimensions. Once again we choose vanishing gaps and $d = 4$. The anomalous dimensions for the choices $\beta_h \in \{-1, 0, 1\}$ are shown in Figure 5. We find that, overall, the dependence on $\beta_h$ is mild, and mostly at the quantitative level. This is a promising indication that, even though off-shell quantities like $\beta$-functions and propagators are inherently gauge-dependent, this dependence is well-controlled. As analytically expected, we find that the two ghost anomalous dimensions agree at $p = 0$ for $\beta_h = 0$. Their value at infinity is also numerically close to their value at zero. As a consequence, for this gauge choice a single, constant ghost anomalous dimension is an accurate approximation. Notably, $\eta_{\text{TT}}$ and $\eta_0$ are also numerically close at $p = 0$ for this choice of gauge.

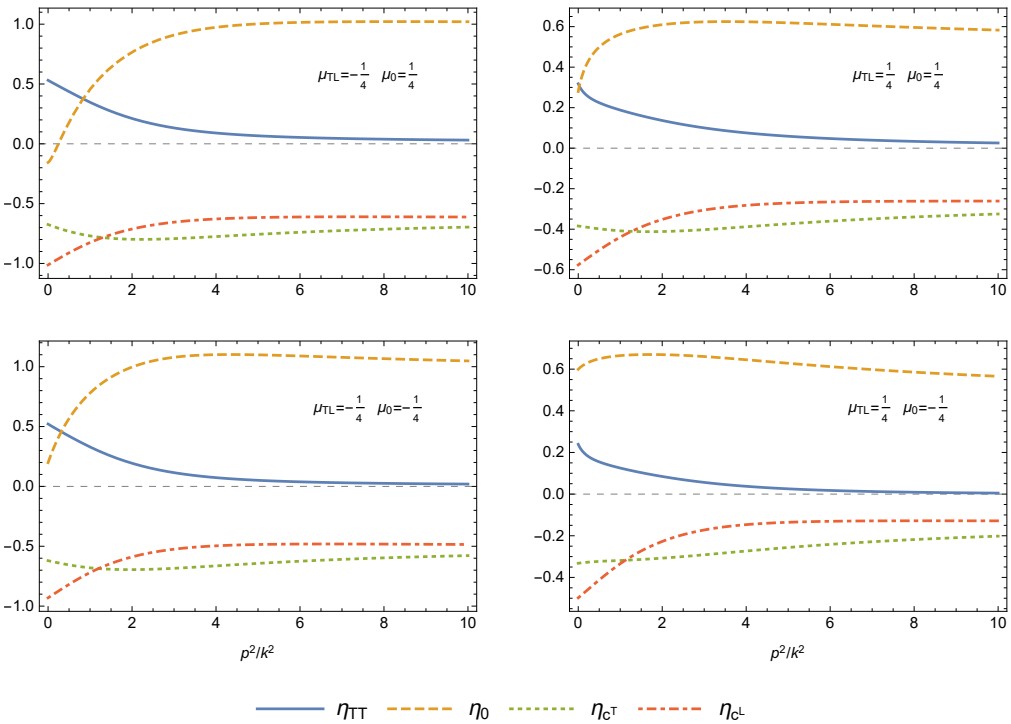

**Figure 4.** Gap dependence of the dynamical anomalous dimensions for the choices $d = 4$, $g = 1$, $\beta_h = 1$ and the regulator (83). For clarity, we have reinstated $k$ in the momentum argument.

### 5.5. Form Factors and the Derivative Expansion

Let us finally discuss the form factors and their derivative expansion obtained via (51). As parameters, we choose $d = 4$ and $\beta_h = 1$, $\mu_{\text{TL}} = \mu_0 = 0$. While this is in general not a fixed point, we want to illustrate the general form of the fluctuation form factors. Since the general shape of the anomalous dimensions seems to be stable under variation of the gaps, see Section 5.3, we expect that the qualitative picture presented here is correct for at least some part of theory space.

The fluctuation form factors, which were reconstructed according to (51), are shown in Figure 6. We find that the $R^2$ form factor $f_R^{\text{fluc}}$ is positive while the $C^2$ form factor $f_C^{\text{fluc}}$ is negative. Due to the positivity of $f_R^{\text{fluc}}$, there is no additional pole in the propagator of the gauge-invariant scalar mode. Similarly, for the transverse-traceless propagator, there is no additional pole. Both go to a constant value for small momenta and fall off with

a power law at large momenta. Due to the absence of additional poles in the graviton propagators, the particle spectrum of the theory at the investigated point agrees with the particle spectrum of GR. Interestingly, the $C^2$ form factor $f_C^{\text{fluc}}$ agrees qualitatively with that obtained in a background approximation of conformally reduced gravity [101].

For small momenta, we find

$$
\begin{aligned}
f_R^{\text{fluc}}(y) &\approx 0.464 + 0.426y - 6.49y^2 + \mathcal{O}(y^3)\,,\\
f_C^{\text{fluc}}(y) &\approx -0.0941 - 0.213y + 3.16y^2 + \mathcal{O}(y^3)\,.
\end{aligned}
\tag{85}
$$

This entails that the worst-case scenario for the derivative expansion explained in Section 3.4 and in [175] is actually realised for this point of theory space: the Taylor coefficients of the two form factors have alternating signs, so a local expansion would not be able to resolve this point accurately. This means that, in any derivative expansion in gravity that is currently technically feasible, viable fixed points might be missed due to the expansion.

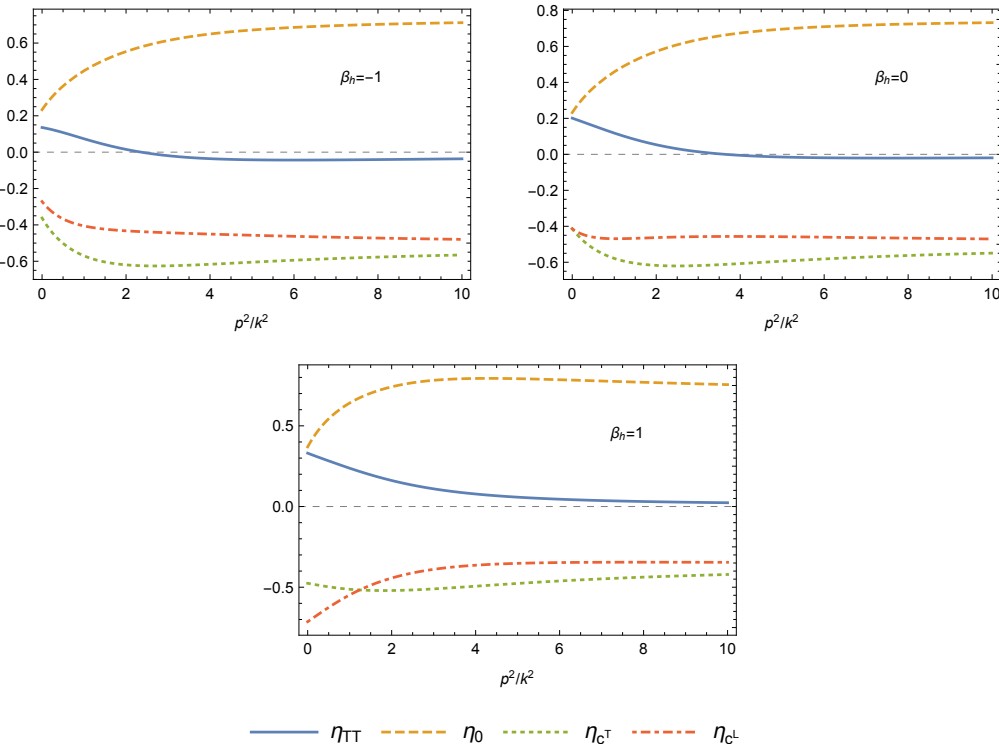

**Figure 5.** Gauge dependence of the dynamical anomalous dimensions for the choices $d = 4$, $g = 1$, $\mu_{\text{TL}} = 0$, $\mu_0 = 0$ and the regulator (83). For clarity, we have reinstated $k$ in the momentum argument.

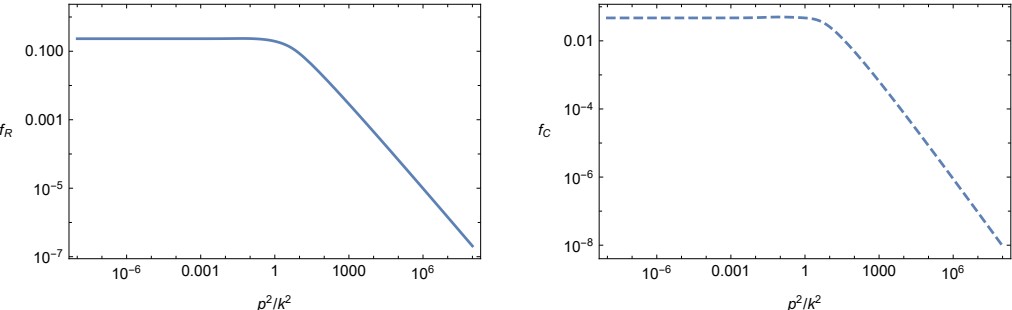

**Figure 6.** Fluctuation form factors reconstructed from the dynamical anomalous dimensions according to (51) for the choices $g = 1, \mu_{TL} = 0, \mu_0 = 0, \beta_h = 1$ and the regulator (83). The dashing indicates that the function is negative. For clarity, we have reinstated $k$ in the momentum argument.

## 6. Momentum-Dependent Background RG Flow

In this section, we provide the results of a computation within the background field approximation. The minimal way to obtain the background form factors is to calculate their forms induced by the Einstein–Hilbert action. For simplicity, we will restrict ourselves to four dimensions and a harmonic gauge fixing, $\alpha_h = \beta_h = 1$. This choice simplifies the calculation so tremendously to justify violating our preference of using the Landau limit. The detailed calculation is shown in Appendix B. In particular, the resulting flow equations are given in Equations (A35)–(A38).

For the form factors, the fixed point equations are linear first order differential equations, so that we can immediately calculate the solution. The free integration constant can be identified with the value of the functions at infinite momentum. With our choice of regulator, at $\Lambda = 0$ we can avoid additional poles in the propagator for

$$f_R(\infty) \gtrsim 2.95, \qquad f_C(\infty) \gtrsim 0.611. \tag{86}$$

To illustrate the solutions, we plot both form factors for two different choices of integration constants of Figures 7 and 8. The first satisfies the above bounds and avoids new poles, whereas the second introduces new poles. We set the cosmological constant to zero and the dimensionless Newton's constant to one. This is, in general, not a fixed point, but allows a more direct comparison with the results of the fluctuation calculation.

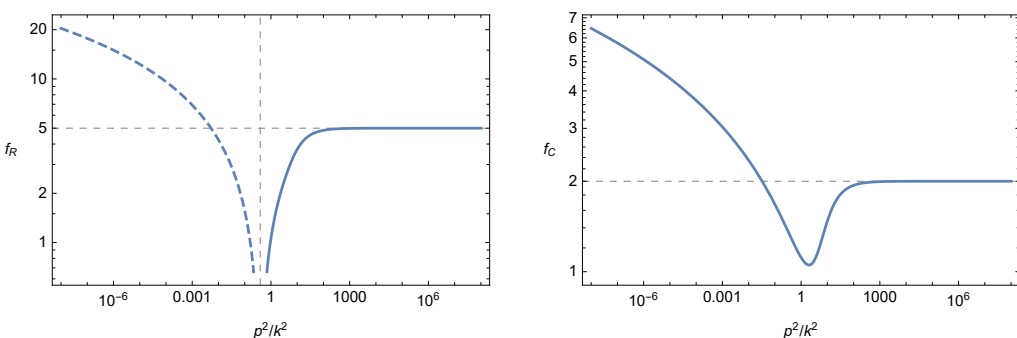

**Figure 7.** Background form factors $f_R$ and $f_C$ for the choices $g = 1, \Lambda = 0, f_R(\infty) = 5, f_C(\infty) = 2$ and the regulator (83). The dashing indicates that the function is negative in that regime. This choice of integration constants avoids additional poles in the propagator. For clarity, we have reinstated $k$ in the momentum argument.

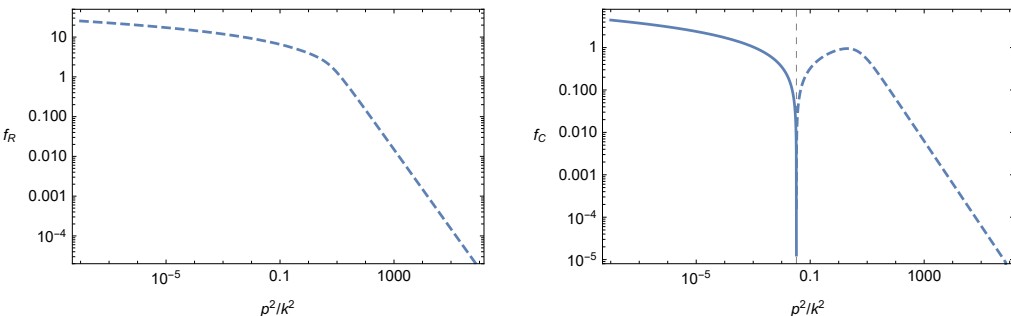

**Figure 8.** Background form factors $f_R$ and $f_C$ for the choices $g = 1, \Lambda = 0, f_R(\infty) = 0, f_C(\infty) = 0$ and the regulator (83). The dashing indicates that the function is negative in that regime. This choice of integration constants introduces additional poles in the propagator. For clarity, we have reinstated $k$ in the momentum argument.

## 7. Comparing Background and Fluctuation Results

We will now compare the results obtained from the background and the fluctuation computation at the level of the form factors in $d = 4$. As our parameters we choose $g = 1, \Lambda = \mu_{\text{TL}} = \mu_0 = 0$. For the background computation, we choose the harmonic gauge $\alpha_h = \beta_h = 1$, whereas for the fluctuation calculation we use $\alpha_h = 0, \beta_h = 1$. The form factors are shown in Figures 6–8.

Since the background computation is essentially one-loop, the form factors show a logarithmic behaviour at small momenta. By contrast, the fluctuation form factors go to finite values at vanishing momentum. The universal one-loop logarithms can be expected to come out correctly once we study the flow in the limit $k \to 0$, at least in some part of theory space where effective field theory around flat spacetime is valid, see also [119].

For large momenta, the background form factors are either finite or zero depending on the choice of integration constant. This is again due to restricting to one-loop. The correct behaviour will be fixed dynamically once the backreaction of these form factors onto the flow is taken into account. The fluctuation form factors show a power law fall-off according to (52). This entails that

$$f_R^{\text{fluc}}(y) \propto y^{-0.85}, \qquad f_C^{\text{fluc}}(y) + \frac{1}{y} \propto y^{-1.17}, \qquad \text{as } y \to \infty. \qquad (87)$$

Regarding additional poles in the propagator, we have to choose the integration constants for the background form factors according to (86) to avoid them. For the fluctuation propagators, no new poles appear at the point that we investigated. The absence of additional poles in the propagators is crucial for the theory to be unitary. At the investigated point, the fluctuation results suggest that indeed no new poles arise, indicating that asymptotically safe quantum gravity might be unitary [120,184–186]. Within the background field approximation, however, the presence of additional poles is not yet conclusive, since in the present setup, they can only be avoided by choosing the integration constant appropriately. Avoiding this choice requires a more consistent computation, which includes the backreaction of the form factors.

Finally, the general shape and the signs differ between the two ways of computation. For the background computation, $f_R$ is negative for small momenta, whereas $f_C$ is positive, in both cases due to the logarithm. For larger momenta, the signs can, but need not change, depending on the integration constant. The fluctuation form factors have a definite sign: positive for $f_R^{\text{fluc}}$ and negative for $f_C^{\text{fluc}}$. They do not show any non-trivial feature besides what is dictated by the asymptotics: they show an approximately constant regime for small momenta, a cross-over at around $p \approx k$, leading to the power law fall-off at large momenta.

It is difficult to predict which of these features are generic in most parts of theory space, and which are special to the point of investigation. We leave this investigation

to future work, together with a detailed analysis of the momentum-dependent flow of interaction vertices.

## 8. Summary and Outlook

In this work, we discussed the full non-perturbative momentum dependence of the graviton and the ghost propagator in quantum gravity. This includes the two different graviton modes with spins zero and two, and the two ghost modes, with spins zero and one. We obtained the results with the help of functional renormalisation group equations, and resolved both gauge and dimensional dependence.

A key result is that the propagators of the different graviton modes behave qualitatively differently. In four dimensions and within our approximation, the spin two mode is momentum-local, whereas the spin zero mode does not show this property. This qualitative result is independent of the gauge choice, and thus potentially has physical significance [31,107]. By contrast, the two ghost modes agree qualitatively and partially even quantitatively, and their anomalous dimensions are approximately momentum-independent.

The dependence on the choice of gauge as well as on the gaps is largely quantitative rather than qualitative. We take this as a sign that the selected examples presented in this work are indeed representative of larger parts of theory space. Furthermore, the weak dependence of the anomalous dimensions on the choice of gauge is a promising indication that, even in truncations, the FRG gives rise to reliable and stable results.

In dimensions larger than four, we find that all anomalous dimensions quickly go to zero. Assuming a regulator that falls off quickly enough, this suppression for large dimensions is exponential in the dimension $d$.

All these results have been obtained from calculations of dynamical correlation functions. To also compare to a computation in a background field approximation, we have calculated the background form factors induced by an Einstein–Hilbert action to quadratic order in the curvature. To facilitate this comparison, we created a dictionary relating the form factors to anomalous dimensions, both evaluated at a fixed point. We find that the one-loop structure of our approximation in the background sector makes such a comparison difficult, since many features are dictated by the one-loop form rather than dynamically, for example, the asymptotic behaviour for both large and small momenta. A dynamical computation of the background form factors with backreaction is desirable but is postponed for the future.

The present work represents a major step in a complete and systematic computation of correlation functions in quantum gravity. The natural next step is to resolve the three-graviton vertex. Since the complete vertex is rather complicated, two possible paths are the resolution of the full momentum dependence of selected tensor structures, or the complete resolution of all tensor structures in a derivative expansion. While the former strategy allows us to extract the full momentum dependence in one specific sector, it is not guaranteed that such an approximation has the desirable features the full theory should have [175]. In the latter strategy, all tensor structures are taken into account; however, as we have seen in Section 5.5, a derivative expansion might not be able to resolve the fixed points of the theory accurately.

Finally, we expect that disentangling the different graviton modes will play an important role in the discussion of spectral functions [119]. In particular, the two sectors will have individual spectral functions, and their different high-momentum behaviour will have a non-trivial effect.

**Supplementary Materials:** The following are available online at https://www.mdpi.com/article/10.3390/universe7070216/s1.

**Author Contributions:** All authors contributed equally to this article. All authors have read and agreed to the published version of the manuscript.

**Funding:** The research of B.K. has been supported by the Perimeter Institute for Theoretical Physics. Research at the Perimeter Institute is supported in part by the Government of Canada through the

Department of Innovation, Science and Economic Development and by the Province of Ontario through the Ministry of Colleges and Universities. M.S. has been supported by the German Academic Scholarship Foundation and gratefully acknowledges hospitality at Syracuse University and at CP3-Origins, University of Southern Denmark, during different stages of this project.

**Data Availability Statement:** The data presented in this study are available in the supplemental notebook.

**Acknowledgments:** We would like to thank Stefan Lippoldt, Alessia Platania and Manuel Reichert for insightful discussions and Alessia Platania and Manuel Reichert for constructive feedback on the manuscript.

**Conflicts of Interest:** The authors declare no conflict of interest. The funders had no role in the design of the study; in the collection, analyses, or interpretation of data; in the writing of the manuscript, or in the decision to publish the results.

## Appendix A. Asymptotic Expansion of Form Factors

In this appendix we derive the asymptotic expansion (52). For this, we consider the integral in the exponent of (51),

$$\Im(y) = \int_0^y \mathrm{d}s \, \frac{\eta_*(s) - \eta_*(0)}{2s} \,. \tag{A1}$$

We will assume that $\eta_*$ is a bounded function on the whole positive real line, which is indeed true in our case. With that, we can first show that the leading order behaviour of (A1) at large $y$ is logarithmic. For that, let us take the derivative of (A1):

$$\Im'(y) = \frac{\eta_*(y) - \eta_*(0)}{2y} \,. \tag{A2}$$

Since $\eta_*$ is bounded, for large $y$, we find that

$$\Im'(y) \sim \frac{\eta_*(\infty) - \eta_*(0)}{2y} \,, \qquad \text{as } y \to \infty \,. \tag{A3}$$

As a consequence, the integral itself is asymptotic to a logarithm,

$$\Im(y) \sim \frac{\eta_*(\infty) - \eta_*(0)}{2} \ln y \,, \qquad \text{as } y \to \infty \,. \tag{A4}$$

We still have to determine the sub-leading behaviour. For that, let us take the difference of the integral and the logarithm above, and take the limit of $y$ going to infinity. We will find that this limit exists, so that the sub-leading part is a constant. To achieve that, let us rewrite the logarithm via

$$\ln y = \left[ \int_0^y \mathrm{d}s \, \frac{1}{1+s} \right] - \ln \frac{1+y}{y} \,. \tag{A5}$$

The virtue of this rewriting is that the logarithm on the right-hand side vanishes in the limit of large $y$. We can then combine the integrals so that

$$\Im(y) - \frac{\eta_*(\infty) - \eta_*(0)}{2} \ln y = \frac{\eta_*(\infty) - \eta_*(0)}{2} \ln \frac{1+y}{y} \\ + \int_0^y \mathrm{d}s \left[ \frac{\eta_*(s) - \eta_*(0)}{2s} - \frac{\eta_*(\infty) - \eta_*(0)}{2(1+s)} \right] \,. \tag{A6}$$

We can now take the limit $y \to \infty$. The integral converges in this limit, since the terms falling off such as $1/s$ at large $s$ cancel, so that finally we get

$$\Im(y) \sim \frac{\eta_*(\infty) - \eta_*(0)}{2} \ln y + \int_0^\infty ds \left[ \frac{\eta_*(s) - \eta_*(0)}{2s} - \frac{\eta_*(\infty) - \eta_*(0)}{2(1+s)} \right], \qquad \text{as } y \to \infty. \tag{A7}$$

This completes the derivation of the asymptotic formula (52).

## Appendix B. Calculation of the Background form Factors Induced by GR

In this appendix we derive the flow equations for $G_N$ and $\Lambda$ as well as the two-curvature form factors induced by the Einstein–Hilbert truncation in a background field approximation and in four dimensions. To make our life easier, we will choose the harmonic gauge condition

$$\alpha_h = \beta_h = 1, \qquad \gamma_h = 0. \tag{A8}$$

Even though we argued in the main text that only the Landau limit allows for a clean flow, in a background field calculation this gauge choice simplifies the calculation tremendously. The reason for this is that in this gauge, the two-point function is diagonal and only depends on Laplacians and curvatures, while it does not involve uncontracted derivatives. Splitting the graviton into traceless and trace parts,

$$h_{\mu\nu} = h_{\mu\nu}^{\text{TL}} + \frac{1}{4} \bar{g}_{\mu\nu} h, \qquad \bar{g}^{\mu\nu} h_{\mu\nu}^{\text{TL}} = 0, \tag{A9}$$

the individual two-point functions at vanishing fluctuation fields read

$$\frac{\delta^2 \Gamma_k}{\delta h^{\text{TL}\mu\nu} \delta h_{\rho\sigma}^{\text{TL}}} \propto \frac{1}{G_N} \left[ \bar{\Delta}_{2\mu\nu}{}^{\rho\sigma} - 2\Lambda \Pi^{\text{TL}}{}_{\mu\nu}{}^{\rho\sigma} \right],$$

$$\frac{\delta^2 \Gamma_k}{\delta h^2} \propto \frac{1}{G_N} [\bar{\Delta} - 2\Lambda], \tag{A10}$$

$$\frac{\delta^2 \Gamma_k}{\delta \bar{c}_\mu \delta c_\nu} \propto \bar{\Delta}_c^{\mu\nu}.$$

Here, we introduced the operators

$$\bar{\Delta}_{2\mu\nu}{}^{\rho\sigma} = \left( \bar{\Delta} - \frac{2}{3} \bar{R} \right) \Pi^{\text{TL}}{}_{\mu\nu}{}^{\rho\sigma} - 2\bar{C}_{(\mu}{}^\rho{}_{\nu)}{}^\sigma,$$

$$\bar{\Delta}_c^{\mu\nu} = \bar{\Delta} \bar{g}^{\mu\nu} - \bar{R}^{\mu\nu}. \tag{A11}$$

These are also the operators that we regularise. This yields the very simple flow

$$\dot{\Gamma}_k \big|_{h=0} = \frac{1}{2} \text{Tr} \left[ \Pi^{\text{TL}} \frac{\left( 2 - \frac{\dot{G}_N}{G_N} \right) \mathcal{R}_k(\bar{\Delta}_2) - 2\bar{\Delta}_2 \mathcal{R}_k'(\bar{\Delta}_2)}{\bar{\Delta}_2 + \mathcal{R}_k(\bar{\Delta}_2) - 2\Lambda} \right]$$

$$+ \frac{1}{2} \text{Tr} \left[ \Pi^{\text{Tr}} \frac{\left( 2 - \frac{\dot{G}_N}{G_N} \right) \mathcal{R}_k(\bar{\Delta}) - 2\bar{\Delta} \mathcal{R}_k'(\bar{\Delta})}{\bar{\Delta} + \mathcal{R}_k(\bar{\Delta}) - 2\Lambda} \right] \tag{A12}$$

$$- \text{Tr} \left[ \frac{2\mathcal{R}_k(\bar{\Delta}_c) - 2\bar{\Delta}_c \mathcal{R}_k'(\bar{\Delta}_c)}{\bar{\Delta}_c + \mathcal{R}_k(\bar{\Delta}_c)} \right].$$

Here, we used the same shape function in all modes. For later convenience, we introduce the shorthands

$$f_h(x) = \frac{\left( 2 - \frac{\dot{G}_N}{G_N} \right) \mathcal{R}_k(x) - 2x \mathcal{R}_k'(x)}{x + \mathcal{R}_k(x) - 2\Lambda}, \qquad f_c(x) = 2 \frac{\mathcal{R}_k(x) - x \mathcal{R}_k'(x)}{x + \mathcal{R}_k(x)}. \tag{A13}$$

In this notation, the flow reads

$$\dot{\Gamma}_k\big|_{h=0} = \frac{1}{2}\mathrm{Tr}[\Pi_{\mathrm{TL}} f_h(\bar{\Delta}_2)] + \frac{1}{2}\mathrm{Tr}[\Pi_{\mathrm{Tr}} f_h(\bar{\Delta})] - \mathrm{Tr} f_c(\bar{\Delta}_c) \equiv \mathcal{T}_{\mathrm{TL}} + \mathcal{T}_{\mathrm{Tr}} + \mathcal{T}_c . \tag{A14}$$

To improve the readability, in the following, we will refrain from using an overbar for background quantities.

*Appendix B.1. Early Time Heat Kernel Expansion*

To evaluate the traces, we will employ the early time expansion of the heat kernel. For an operator of Laplace type $\Delta + \mathbb{E}$, where $\mathbb{E}$ is an endomorphism with the appropriate bundle structure, the expansion reads [187]

$$\mathrm{Tr}\, e^{-s(\Delta + \mathbb{E})} \simeq \frac{1}{(4\pi s)^{d/2}} \int \mathrm{d}^d x \, \sqrt{g} \, \mathrm{tr}\bigg\{ \mathbb{1} - s\mathbb{E} + \frac{s}{6} R\, \mathbb{1} + s^2 \Big[ \mathbb{1} \, R_{\mu\nu} \, f_{Ric}(s\Delta) \, R^{\mu\nu}$$

$$+ \mathbb{1} \, R \, f_R(s\Delta) \, R + R \, f_{RE}(s\Delta) \, \mathbb{E} + \mathbb{E} \, f_E(s\Delta) \, \mathbb{E} + \mathcal{F}_{\mu\nu} \, f_F(s\Delta) \, \mathcal{F}^{\mu\nu} \Big] \bigg\} . \tag{A15}$$

In this expression, Tr is a functional trace, tr is the trace over the bundle indices, $\mathcal{F}$ is the bundle curvature

$$\mathcal{F}_{\mu\nu} = [D_\mu, D_\nu] , \tag{A16}$$

which depends on the index structure of the field traced over, and

$$f_{Ric}(x) = \frac{f(x) - 1 + \frac{x}{6}}{x^2} , \tag{A17}$$

$$f_R(x) = \frac{f(x)}{32} + \frac{f(x) - 1}{8x} - \frac{f(x) - 1 + \frac{x}{6}}{8x^2} , \tag{A18}$$

$$f_{RE}(x) = -\frac{f(x)}{4} - \frac{f(x) - 1}{2x} , \tag{A19}$$

$$f_E(x) = \frac{f(x)}{2} , \tag{A20}$$

$$f_F(x) = -\frac{f(x) - 1}{2x} , \tag{A21}$$

are heat kernel form factors. The universal heat kernel function $f$ is defined by

$$f(x) = \int_0^1 \mathrm{d}\xi \, e^{-x\xi(1-\xi)} = 2\int_0^{\frac{1}{2}} \mathrm{d}\xi \, e^{-x\xi(1-\xi)} = \sqrt{\frac{\pi}{x}} e^{-\frac{x}{4}} \, \mathrm{erfi}\!\left(\frac{\sqrt{x}}{2}\right) = 1 - \frac{x}{6} + \mathcal{O}(x^2) . \tag{A22}$$

The Taylor expansion shows that all form factors are regular at zero.

To bring the traces (A12) into the form of the standard heat kernel trace (A15), we use the inverse Laplace transform. For a function $g$ of an operator $\mathcal{O}$, this entails that we can write

$$g(\mathcal{O}) = \int_0^\infty \mathrm{d}s \, \tilde{g}(s) \, e^{-s\mathcal{O}} . \tag{A23}$$

To be in line with the basis of the main text, we can use the relation

$$\int \mathrm{d}^d x \, \big[ R_{\mu\nu} \, \Delta^n \, R^{\mu\nu} \big] = \int \mathrm{d}^d x \, \left[ \frac{1}{3} R \, \Delta^n \, R + \frac{1}{2} C_{\mu\nu\rho\sigma} \, \Delta^n \, C^{\mu\nu\rho\sigma} \right] + \mathcal{O}(\mathcal{R}^3) , \qquad n > 0 , \tag{A24}$$

to replace any occurrence of Ricci tensors by Ricci scalars and Weyl tensors. In the following, we will also neglect the Euler characteristic, which corresponds to the case $n = 0$ in the above equation.

*Appendix B.2. Some Integral Transformations in d = 4*

To bring the flow equations into a convenient form, we will use the formula

$$\int_0^\infty ds\, \tilde{g}(s) s^{-n} = \frac{1}{\Gamma(n)} \int_0^\infty dz\, z^{n-1} g(z),$$ (A25)

which holds for $n > 0$. Also, the following identities hold true:

$$\int_0^\infty ds\, \tilde{g}(s)\, f(s\Delta) = 2 \int_0^{\frac{1}{4}} du\, \frac{1}{\sqrt{1-4u}}\, g(u\,\Delta),$$ (A26)

$$\int_0^\infty ds\, \tilde{g}(s)\, \frac{f(s\Delta)-1}{s\Delta} = -\int_0^{\frac{1}{4}} du\, \sqrt{1-4u}\, g(u\,\Delta),$$ (A27)

$$\int_0^\infty ds\, \tilde{g}(s)\, \frac{f(s\Delta)-1+\frac{s\Delta}{6}}{s^2\Delta^2} = \frac{1}{6} \int_0^{\frac{1}{4}} du\, (1-4u)^{3/2} g(u\,\Delta).$$ (A28)

In all these equations, $\tilde{g}$ is the inverse Laplace transform of an arbitrary function $g$, see (A23), and $f$ is the universal heat kernel function (A22). The equations can be shown by inserting the integral form of $f$ and performing repeated variable transformations while exchanging the order of integrals.

*Appendix B.3. Trace Contribution of the Spin Zero Part*

We start with the contribution of the trace part of the graviton to the flow. The trace projector makes it a scalar trace, with

$$\mathcal{F}_{\mu\nu} = 0, \qquad \mathbb{E} = 0.$$ (A29)

Using the inverse Laplace transform (A23), then applying the general trace formula (A15), and using the integral identities (A25)–(A28) for this case and in $d = 4$ gives

$$
\begin{aligned}
\mathcal{T}_{\text{Tr}} \simeq \frac{1}{2} \frac{1}{16\pi^2} \int d^4x\, \sqrt{g} \Bigg\{ & \int_0^\infty dz \left[ z f_h(z) + \frac{1}{6} R f_h(z) \right] \\
& + \int_0^{\frac{1}{4}} du \left[ \frac{1}{12} C_{\mu\nu\rho\sigma}(1-4u)^{3/2} f_h(u\,\Delta) C^{\mu\nu\rho\sigma} \right. \\
& \left. + R \left\{ \frac{1}{16} \frac{1}{\sqrt{1-4u}} - \frac{1}{8}\sqrt{1-4u} + \frac{5}{144}(1-4u)^{3/2} \right\} f_h(u\,\Delta) R \right] \Bigg\}.
\end{aligned}
$$ (A30)

*Appendix B.4. Trace Contribution of the Ghost*

Next, we present the contribution of the ghost. In this case,

$$\mathbb{E}_{\mu\nu} = -R_{\mu\nu}, \qquad \text{tr}\mathcal{F}_{\mu\nu}\, g(\Delta)\, \mathcal{F}^{\mu\nu} = -R_{\mu\nu\rho\sigma}\, g(\Delta)\, R^{\mu\nu\rho\sigma}.$$ (A31)

With this, the trace is

$$
\begin{aligned}
\mathcal{T}_c \simeq -\frac{1}{16\pi^2} \int d^4x\, \sqrt{g} \Bigg\{ & \int_0^\infty dz \left[ 4z f_c(z) + \frac{5}{3} R f_c(z) \right] \\
& + \int_0^{\frac{1}{4}} du \left[ C_{\mu\nu\rho\sigma} \left\{ \frac{1}{2} \frac{1}{\sqrt{1-4u}} - \sqrt{1-4u} + \frac{1}{3}(1-4u)^{3/2} \right\} f_c(u\,\Delta) C^{\mu\nu\rho\sigma} \right. \\
& \left. + R \left\{ \frac{13}{12} \frac{1}{\sqrt{1-4u}} - \frac{7}{6}\sqrt{1-4u} + \frac{5}{36}(1-4u)^{3/2} \right\} f_c(u\,\Delta) R \right] \Bigg\}.
\end{aligned}
$$ (A32)

*Appendix B.5. Trace Contribution of the Spin Two Part*

Finally, we present the contribution of the traceless spin two component of the graviton. With

$$\mathbb{E}_{\mu\nu\rho\sigma} = \frac{2}{3} R \, \Pi_{\text{TL}\mu\nu\rho\sigma} - C_{\mu\rho\nu\sigma} - C_{\nu\rho\mu\sigma} \,,$$

$$\text{tr}\Pi_{\text{TL}}\mathcal{F}_{\mu\nu}\, g(\Delta) \,\mathcal{F}^{\mu\nu} \simeq -6 C_{\mu\nu\rho\sigma}\, g(\Delta)\, C^{\mu\nu\rho\sigma} + 2R\, g(\Delta)\, R - 12 R_{\mu\nu}\, g(\Delta)\, R^{\mu\nu} \,, \tag{A33}$$

we find

$$\mathcal{T}_{\text{TL}} \simeq \frac{1}{2}\frac{1}{16\pi^2} \int \mathrm{d}^4 x \,\sqrt{g} \left\{ \int_0^\infty \mathrm{d}z \left[ 9z f_h(z) - \frac{9}{2} R f_h(z) \right] \right.$$

$$+ \int_0^{\frac{1}{4}} \mathrm{d}u \left[ C_{\mu\nu\rho\sigma} \left\{ 3\frac{1}{\sqrt{1-4u}} - 6\sqrt{1-4u} + \frac{3}{4}(1-4u)^{3/2} \right\} f_h(u\,\Delta) C^{\mu\nu\rho\sigma} \right. \tag{A34}$$

$$\left. \left. + R \left\{ \frac{25}{16}\frac{1}{\sqrt{1-4u}} + \frac{7}{8}\sqrt{1-4u} + \frac{5}{16}(1-4u)^{3/2} \right\} f_h(u\,\Delta) R \right] \right\} \,.$$

*Appendix B.6. Background Flow Equations*

Now that we have computed all traces, we can read off the beta functions of $G_N$, $\Lambda$ and the two-curvature form factors by comparing the coefficients of the scale derivative acting on the action (40), with the results for (A14), given in (A30), (A32) and (A34). We will present them for the dimensionless quantities $g$, $\lambda$, $f_R$ and $f_c$. They read

$$\dot{g} = g \frac{2 - \frac{g}{3\pi} \int_0^\infty \mathrm{d}z \left[ 13\frac{2\mathcal{R}_k(z) - z\mathcal{R}_k'(z)}{z + \mathcal{R}_k(z) - 2\lambda} + 10\frac{\mathcal{R}_k(z) - z\mathcal{R}_k'(z)}{z + \mathcal{R}_k(z)} \right]}{1 - \frac{13g}{6\pi} \int_0^\infty \mathrm{d}z \frac{\mathcal{R}_k(z)}{z + \mathcal{R}_k(z) - 2\lambda}} \,, \tag{A35}$$

$$\dot{\lambda} = \left( -4 + \frac{\dot{g}}{g} \right)\lambda - \frac{5\dot{g}}{2\pi} \int_0^\infty \mathrm{d}z\, z\, \frac{\mathcal{R}_k(z)}{z + \mathcal{R}_k(z) - 2\lambda}$$

$$+ \frac{g}{\pi} \int_0^\infty \mathrm{d}z\, z \left[ 5\frac{2\mathcal{R}_k(z) - z\mathcal{R}_k'(z)}{z + \mathcal{R}_k(z) - 2\lambda} - 4\frac{\mathcal{R}_k(z) - z\mathcal{R}_k'(z)}{z + \mathcal{R}_k(z)} \right] \,, \tag{A36}$$

$$\dot{f}_R(z) = \frac{\dot{g}}{g} f_R(z) + 2z\, f_R'(z) + \frac{3\dot{g}}{\pi} \int_0^{\frac{1}{4}} \mathrm{d}u\, \mu_h^R(u) \frac{\mathcal{R}_k(u\,z)}{u\,z + \mathcal{R}_k(u\,z) - 2\lambda}$$

$$- \frac{6g}{\pi} \int_0^{\frac{1}{4}} \mathrm{d}u \left[ \mu_h^R(u) \frac{2\mathcal{R}_k(u\,z) - u\,z\mathcal{R}_k'(u\,z)}{u\,z + \mathcal{R}_k(u\,z) - 2\lambda} + \mu_c^R(u) \frac{\mathcal{R}_k(u\,z) - u\,z\mathcal{R}_k'(u\,z)}{u\,z + \mathcal{R}_k(u\,z)} \right] \,, \tag{A37}$$

$$\dot{f}_C(z) = \frac{\dot{g}}{g} f_C(z) + 2z\, f_C'(z) - \frac{\dot{g}}{\pi} \int_0^{\frac{1}{4}} \mathrm{d}u\, \mu_h^C(u) \frac{\mathcal{R}_k(u\,z)}{u\,z + \mathcal{R}_k(u\,z) - 2\lambda}$$

$$+ \frac{2g}{\pi} \int_0^{\frac{1}{4}} \mathrm{d}u \left[ \mu_h^C(u) \frac{2\mathcal{R}_k(u\,z) - u\,z\mathcal{R}_k'(u\,z)}{u\,z + \mathcal{R}_k(u\,z) - 2\lambda} + \mu_c^C(u) \frac{\mathcal{R}_k(u\,z) - u\,z\mathcal{R}_k'(u\,z)}{u\,z + \mathcal{R}_k(u\,z)} \right] \,. \tag{A38}$$

Here, we introduced the combined measures

$$\mu_h^R(u) = \frac{13}{8}\frac{1}{\sqrt{1-4u}} + \frac{3}{4}\sqrt{1-4u} + \frac{25}{72}(1-4u)^{3/2} \,,$$

$$\mu_c^R(u) = -\frac{13}{6}\frac{1}{\sqrt{1-4u}} + \frac{7}{3}\sqrt{1-4u} - \frac{5}{18}(1-4u)^{3/2} \,,$$

$$\mu_h^C(u) = 3\frac{1}{\sqrt{1-4u}} - 6\sqrt{1-4u} + \frac{5}{6}(1-4u)^{3/2} \,, \tag{A39}$$

$$\mu_c^C(u) = -\frac{1}{\sqrt{1-4u}} + 2\sqrt{1-4u} - \frac{2}{3}(1-4u)^{3/2} \,.$$

*Appendix B.7. Fixed Point Structure*

Let us now analyse the fixed point structure of these background flow equations. Employing the fixed point condition

$$\dot{g} = \dot{\lambda} = \dot{f}_R = \dot{f}_C = 0, \tag{A40}$$

we find that the form factor equations are first order linear ordinary differential equations of the form

$$2z\, f'(z) = \int_0^{\frac{1}{4}} \mathrm{d}u\, \mu(u)\mathcal{K}(u\,z), \tag{A41}$$

for some kernel $\mathcal{K}$. The explicit solution to this equation reads

$$f(\infty) - f(\Delta) = \int_0^{\frac{\Delta}{4}} \mathrm{d}x \int_\Delta^\infty \mathrm{d}z\, \frac{1}{2z^2}\, \mu\left(\frac{x}{z}\right) \mathcal{K}(x) + \int_{\frac{\Delta}{4}}^\infty \mathrm{d}x \int_{4x}^\infty \mathrm{d}z\, \frac{1}{2z^2}\, \mu\left(\frac{x}{z}\right) \mathcal{K}(x). \tag{A42}$$

We chose to impose the boundary condition at $\Delta = \infty$ since for this choice the integrals on the right-hand side converge for all $\Delta \geq 0$. Inserting the explicit expressions for the kernel, we can also perform the $z$-integration. With the shorthand notation

$$\nu(a,b,c|\omega) = a\left(1 - (1-\omega)^{1/2}\right) + \frac{b}{3}\left(1 - (1-\omega)^{3/2}\right) + \frac{c}{5}\left(1 - (1-\omega)^{5/2}\right), \tag{A43}$$

the explicit induced background fixed point form factors are

$$
\begin{aligned}
f_R(\Delta) = f_R(\infty) - \frac{3g}{2\pi}\Bigg[ & \int_0^1 \frac{\mathrm{d}\omega}{\omega}\, \nu\left(\frac{13}{8},\frac{3}{4},\frac{25}{72}\,\bigg|\,\omega\right) \frac{2\mathcal{R}_k(\frac{\omega\Delta}{4}) - \frac{\omega\Delta}{4}\mathcal{R}_k'(\frac{\omega\Delta}{4})}{\frac{\omega\Delta}{4} + \mathcal{R}_k(\frac{\omega\Delta}{4}) - 2\lambda} \\
& + \int_1^\infty \frac{\mathrm{d}\omega}{\omega}\, \nu\left(\frac{13}{8},\frac{3}{4},\frac{25}{72}\,\bigg|\,1\right) \frac{2\mathcal{R}_k(\frac{\omega\Delta}{4}) - \frac{\omega\Delta}{4}\mathcal{R}_k'(\frac{\omega\Delta}{4})}{\frac{\omega\Delta}{4} + \mathcal{R}_k(\frac{\omega\Delta}{4}) - 2\lambda} \\
& + \int_0^1 \frac{\mathrm{d}\omega}{\omega}\, \nu\left(-\frac{13}{6},\frac{7}{3},-\frac{5}{18}\,\bigg|\,\omega\right) \frac{\mathcal{R}_k(\frac{\omega\Delta}{4}) - \frac{\omega\Delta}{4}\mathcal{R}_k'(\frac{\omega\Delta}{4})}{\frac{\omega\Delta}{4} + \mathcal{R}_k(\frac{\omega\Delta}{4})} \\
& + \int_1^\infty \frac{\mathrm{d}\omega}{\omega}\, \nu\left(-\frac{13}{6},\frac{7}{3},-\frac{5}{18}\,\bigg|\,1\right) \frac{\mathcal{R}_k(\frac{\omega\Delta}{4}) - \frac{\omega\Delta}{4}\mathcal{R}_k'(\frac{\omega\Delta}{4})}{\frac{\omega\Delta}{4} + \mathcal{R}_k(\frac{\omega\Delta}{4})} \Bigg],
\end{aligned} \tag{A44}
$$

$$
\begin{aligned}
f_C(\Delta) = f_C(\infty) + \frac{g}{2\pi}\Bigg[ & \int_0^1 \frac{\mathrm{d}\omega}{\omega}\, \nu\left(3,-6,\frac{5}{6}\,\bigg|\,\omega\right) \frac{2\mathcal{R}_k(\frac{\omega\Delta}{4}) - \frac{\omega\Delta}{4}\mathcal{R}_k'(\frac{\omega\Delta}{4})}{\frac{\omega\Delta}{4} + \mathcal{R}_k(\frac{\omega\Delta}{4}) - 2\lambda} \\
& + \int_1^\infty \frac{\mathrm{d}\omega}{\omega}\, \nu\left(3,-6,\frac{5}{6}\,\bigg|\,1\right) \frac{2\mathcal{R}_k(\frac{\omega\Delta}{4}) - \frac{\omega\Delta}{4}\mathcal{R}_k'(\frac{\omega\Delta}{4})}{\frac{\omega\Delta}{4} + \mathcal{R}_k(\frac{\omega\Delta}{4}) - 2\lambda} \\
& + \int_0^1 \frac{\mathrm{d}\omega}{\omega}\, \nu\left(-1,2,-\frac{2}{3}\,\bigg|\,\omega\right) \frac{\mathcal{R}_k(\frac{\omega\Delta}{4}) - \frac{\omega\Delta}{4}\mathcal{R}_k'(\frac{\omega\Delta}{4})}{\frac{\omega\Delta}{4} + \mathcal{R}_k(\frac{\omega\Delta}{4})} \\
& + \int_1^\infty \frac{\mathrm{d}\omega}{\omega}\, \nu\left(-1,2,-\frac{2}{3}\,\bigg|\,1\right) \frac{\mathcal{R}_k(\frac{\omega\Delta}{4}) - \frac{\omega\Delta}{4}\mathcal{R}_k'(\frac{\omega\Delta}{4})}{\frac{\omega\Delta}{4} + \mathcal{R}_k(\frac{\omega\Delta}{4})} \Bigg].
\end{aligned} \tag{A45}
$$

Some of the integrals can be performed analytically:

$$\int_1^\infty \frac{\mathrm{d}\omega}{\omega}\, \nu\left(a,b,c\,\bigg|\,1\right) \frac{\mathcal{R}_k(\frac{\omega\Delta}{4}) - \frac{\omega\Delta}{4}\mathcal{R}_k'(\frac{\omega\Delta}{4})}{\frac{\omega\Delta}{4} + \mathcal{R}_k(\frac{\omega\Delta}{4})} = \left(a + \frac{b}{3} + \frac{c}{5}\right) \ln\left(1 + \frac{\mathcal{R}_k(\frac{\Delta}{4})}{\frac{\Delta}{4}}\right), \tag{A46}$$

while the others have to be evaluated numerically.

## Notes

[1]    We refer to $z(y)$ as the dimensionless wave function renormalisation, since its canonical mass dimension and the anomalous dimension cancel at the fixed point.

[2]    A correlation function is called momentum-local in this context if the ratio of its flow to the correlator itself vanishes for large momenta.

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
