# Peer review of "Non-Perturbative Propagators in Quantum Gravity"

_universe, doi:10.3390/universe7070216_

Round 1

Reviewer 1 Report

This paper gives a thorough discussion of the two-point functions of the gravitons and of the ghosts, separating the irreducible components of the fields with different spins.
The analysis is performed in the context of the functional renormalization group, which means that the quantum effects determine first the derivative of the two-point functions with respect to an external cutoff scale. From this information, supplemented by appropriate boundary conditions, one can then reconstruct the form-factors appearing in the terms quadratic in curvature in the effective action.
The paper is generally well written and sufficiently clear and can be accepted for publication. 

Author Response

We thank the referee for their time to review our article, and are happy about their positive evaluation.

Reviewer 2 Report

The paper is concentrated on computing propagators in quantum gravity using the functional renormalization group technique. The paper focuses on the technical advances but seems to lack a clear physical understanding of the results. I think that there are many physical points that need to be expanded on so that the reader gets some indication of whether there is some physical significance to the results. Furthermore, there are some rather arbitrary choices, without any strong justification, which could influence the results and thus prevent any strong conclusions from being drawn.   I would suggest that the authors address the following points, therefore, before the paper can be accepted.

1) One of the main findings is that the spin two modes is “momentum local” while the spin 0 mode is not. First, it is not clear  what the physical consequence of being momentum local is. Is there some connection to locality in real space? What are the consequences if momentum locality is not realised for one of the physical fluctuations? In particular, would the absence of momentum locality mean that recovering classical general relativity in a particular limit or, could it be just a physical characteristic of a consistent theory of quantum gravity, leading to some interesting new physics? Furthermore, is there any indication that momentum locality (or the lack of) is an artefact of the approximation? It would seem that it is, since the authors state that they have to identify g_3 and g_4 to recover momentum locality. Surely it would be remarkable if when calculated g_3 and g_4 came out identical. Maybe there is some symmetry that would imply that this would actually be the case?

2) It is also not clear which physical degrees of freedom are present. In general relativity, based on the Einstein-Hilbert action, there is a spin two mode the graviton, while in higher derivative gravity there can be more physical degrees of freedom. I think that the authors need to clarify which is the particle content that they observe in their propagators. For example, is the spin zero mode related to a scalar particle as in f(R) gravity or is this not a propagating mode like in general relativity? Perhaps the absence of momentum locality in the spin 0 mode is not physical in the latter case. I suppose the choice of the seed action (5) has some bearing on this question.

3) The authors have used one regulator function and fixed g=1 in their numerical calculations.

Both seem to be largely arbitrary choices and thus the possible impact of the results is quite diminished by only considering this point of investigation. For example the alternating signs found in

the derivative expansion (85) could just be a artifact of these choices. I appreciate that the authors have made significant technical advances, however I feel that any significance to their results is questionable without some further investigation. In particular the value of g at a fixed point will depend on the regulator generically, so fixing both g and the form of the regulator independently of each other prevents one from drawing strong conclusions.     

I therefore would like to see at least one of the following studies

I) Vary the form of the regulator. A simple modification would be to multiply (83)  their choice by a parameter A and vary A.

II) Vary the value of g.

III) Close the system of equations for the chosen form of the regulator (83) to determine a value of g at a fixed point rather than setting it to one.   

I also have a technical query

4) The last term in (6) is non-local. Is this non-locality unavoidable if, as the authors claim, it allows one to compute the scale dependence of all tensor structures for the two-point function? Also is the non-locality harmless? My guess is that, if the seed action was to include higher derivative terms, then the non-local gauge fixing would not be necessary.

Author Response

We thank the referee for their time to review our article, and for providing useful and constructive comments.  

Below we reply to the individual points raised by the referee.

1) The concept of momentum locality was introduced in references [29,31,39], where possible physical implications are discussed. In brief, a momentum-local flow might allow to exchange the limits k->0 and p->\infty. This is technically desirable, since it allows to interpret the fixed points discovered by investigating FRG flows in the light of physical RG flows. However, this is just a technical shortcut. If the flows are not momentum-local, one has to take the k->0 limit carefully, and then investigate the large momentum behaviour in this limit. This again highlights the importance of resolving momentum dependence in asymptotically safe quantum gravity.
Indeed, our findings suggest that, unless miraculous cancellations appear, momentum locality might only hold in certain approximations, for example g_3=g_4. Going one step further, the scale dependence of g_3 and g_4 will depend on g_4, g_5, and g_5, g_6, respectively. It is to be expected that momentum locality for these higher-order correlation functions also only holds for some specific relation of the couplings.
As a matter of fact, reference [39] computed the fixed-point values of g_3 and g_4 within some approximation, and a semi-quantitative agreement was found. This semi-quantitative agreement between different n-point correlators was also discovered in gravity-matter systems and was named effective universality (references [64-66]). In analogy to QCD, where different n-point correlators related to the strong coupling agree in the perturbative regime, it was interpreted as an indication for a near-perturbative nature of quantum gravity. We have added a paragraph below Eq. (65) drawing the connection to references [39] as well as [64-66].

2) At the investigated point in the gravitational parameter space, we do not encounter additional poles in the graviton propagator. Therefore, our system has the same particle spectrum as GR. We have added a comment on this before Eq. (85), where we discuss the reconstructed form factors.

3) We agree that the choices for g and the regulator are very specific and to some extent arbitrary. However, without closing the system with at least one three-point function, and computing the fixed point of the system, any value of g is as arbitrary as any other point. Closing the system of equations is however well beyond the scope of the present work, since it requires to compute and feed back the scale dependence of at least the three-point vertex. Of course, an extensive and systematic investigation of the dependence on the regulator, as well as on g, would be desirable to find out which regions in the gravitational space feature additional poles, but this goes beyond the scope of the paper, and moreover would provide only limited insight without closing the flow equations.

For small g, the anomalous dimensions are dominated by the quantum term which is not proportional to the anomalous dimensions themselves. In that way, the anomalous dimensions have a controlled expansion in powers of g, multiplied by nested integrals. In that sense, the value g=1 is already rather non-perturbative. Considering that typical fixed point values in the (fluctuation computation) literature are smaller than g=1, we are confident that the most interesting part of theory space behaves qualitatively the same as our selected examples. A short comment has been added on that at the end of section 5.1.

4) The non-locality in Eq. (6) is only introduced to avoid a dimensionful gauge parameter \gamma_h. One could easily avoid this tensor structure by making \gamma_h dimensionful, and in the same way compute the scale dependence of this parameter.
Furthermore, since \gamma_h can be shifted to zero by adjusting \beta_h, on the level of the flow equations, this non-locality is harmless. It does not enter the flow of physical n-point correlators.

We thank the referee again for their very useful comments that have helped us to improve our paper. We hope that we could clarify all issues.

Reviewer 3 Report

In this paper, using the non-perturbative renormalisation group methods, the authors compute the full momentum
dependence of propagators in quantum gravity in general dimensions. They successfully distinguish between all the different
modes of the gravitons and the ghosts. Therefore, many interesting results are obtained, for example, the propagators of the two ghost modes related for asymptotic momenta, but
differ for finite momenta; the quantum corrections to the free propagator decreasing exponentially with increasing
dimension and so on. These results may be important for the research in this field. In all, the paper is very well written and I am glad to recommend its publication.   

Author Response

(The authors gave the same response as above.)

Round 2

Reviewer 2 Report

The authors have addressed my concerns and I'm happy with the improvements to the paper. I recommend the paper be published.